# Multi-Marginal Stochastic Flow Matching
# for High-Dimensional Snapshot Data at Irregular Time Points

**Justin Lee** [1]  **Behnaz Moradijamei** [2]  **Heman Shakeri** [1]

## Abstract

Modeling the evolution of high-dimensional systems from limited snapshot observations at irregular time points poses a significant challenge in quantitative biology and related fields. Traditional approaches often rely on dimensionality reduction techniques, which can oversimplify the dynamics and fail to capture critical transient behaviors in non-equilibrium systems. We present Multi-Marginal Stochastic Flow Matching (MMSFM), a novel extension of simulation-free score and flow matching methods to the multi-marginal setting, enabling the alignment of high-dimensional data measured at non-equidistant time points without reducing dimensionality. The use of measure-valued splines enhances robustness to irregular snapshot timing, and score matching prevents overfitting in high-dimensional spaces. We validate our framework on several synthetic and benchmark datasets, including gene expression data collected at uneven time points and an image progression task, demonstrating the method's versatility.[1]

## 1. Introduction

Understanding cellular responses to perturbations is a fundamental challenge in quantitative biology, with significant implications for fields such as developmental biology, cancer research, and drug discovery (Altschuler & Wu, 2010; Saeys et al., 2016). Modeling these responses requires capturing complex stochastic dynamics in high-dimensional cellular states that evolve over time under the influence of both deterministic and random factors. Developing generative models that accurately represent these dynamics is crucial for simulating cellular behavior and predicting responses in unseen cells under non-steady state dynamics imposed by perturbation agents such as drug stimuli.

A common approach to modeling such systems is through stochastic differential equations (SDEs), particularly the Langevin equation as an Itô SDE (Gardiner, 1985; Risken & Risken, 1996); the evolution of the cellular state $X(t) \in \mathbb{R}^d$ can be described by

$$dX(t) = u_t(X(t)) \, dt + g(t) \, dW(t), \qquad (1)$$

where $u_t(x)$ is the drift term representing deterministic dynamics, $g(t)$ is the diffusion coefficient capturing stochastic fluctuations, and $W(t)$ is a Wiener process modeling random noise. At the population level, the corresponding probability density function $p(x, t)$ evolves according to the Fokker-Planck equation (Risken & Risken, 1996):

$$\frac{\partial p_t(x)}{\partial t} = -\nabla \cdot (p_t(x) \, u_t(x)) + \frac{g^2(t)}{2} \, \Delta p_t(x), \quad (2)$$

where $p_t(x)$ is shorthand for $p(x, t)$, $\nabla \cdot$ denotes the divergence operator, and $\Delta$ is the Laplacian operator.

In practice we only observe the system through snapshot measurements at discrete, possibly irregular time points $t_0 < t_1 < \cdots < t_M$, providing samples from the marginal distributions $\rho_i = p_{t_i}(x)$ (Schofield et al., 2023). Therefore, we lack trajectory data that would reveal how individual states evolve between these snapshots due to the destructive nature of single-cell measurements. This raises a fundamental question: *among the infinitely many stochastic processes that could connect these observed marginals (Weinreb et al., 2018), which one is the most likely?*

### 1.1. Least Action Principle

To address this problem, we turn to the theory of *Optimal Transport* (OT) (Villani, 2009) which seeks the most efficient way to transform one probability distribution into another. In the simplest case of two marginals $\rho_0$ and $\rho_1$, OT aims to find a transport map $T$ that minimizes the cost

---

[1]School of Data Science, University of Virginia, Charlottesville VA, USA. [2]James Madison University, Harrisonburg VA, USA. Correspondence to: Justin Lee <jgh2xh@virginia.edu>, Heman Shakeri <hs9hd@virginia.edu>.

*Proceedings of the 42nd International Conference on Machine Learning*, Vancouver, Canada. PMLR 267, 2025. Copyright 2025 by the author(s).

[1]Code available at https://github.com/Shakeri-Lab/MMSFM

functional:

$$\min_{T} \int \|x - T(x)\|^2 \, d\rho_0(x) \quad \text{subject to } T_{\#}\rho_0 = \rho_1, \quad (3)$$

where $T_{\#}\rho_0$ denotes the pushforward of $\rho_0$ under $T$. Kantorovich's generalized formulation of (3) is a linear programming problem over the set of joint probability distributions, leading to the definition of the Wasserstein-2 distance:

$$W_2^2(\rho_0, \rho_1) = \min_{\pi \in \Pi(\rho_0, \rho_1)} \int \|x - y\|^2 \, d\pi(x, y), \quad (4)$$

where $\Pi(\rho_0, \rho_1)$ is the set of joint distributions with marginals $\rho_0$ and $\rho_1$. While OT provides a deterministic model based on the principle of least action—finding the shortest path or geodesic in the space of probability distributions—it does not account for the inherent stochasticity of biological systems (Horowitz & Gingrich, 2020). Cells are subject to both extrinsic noise, such as variations in initial conditions and environmental inputs (Hilfinger & Paulsson, 2011), and intrinsic noise arising from the thermodynamic uncertainty in biochemical reactions (Mitchell & Hoffmann, 2018).

To incorporate stochasticity and identify the most likely stochastic process connecting the observed marginals, we consider the entropic-regularized optimal transport problem, a particular case of the *Schrödinger Bridge Problem* (SBP) (Schrödinger, 1931; Léonard, 2014). The SBP seeks the stochastic process that minimally deviates from a prior—typically a Brownian motion—while matching the observed marginals. It can be considered a general statistical inference and model improvement methodology in which one updates the probability of a hypothesis based on the most recent observations while making the fewest possible assumptions beyond the available information (Pavon et al., 2021). This approach aligns with Occam's razor principle and aims to find the simplest stochastic process that explains the data with minimal adjustment to our prior belief.

**Extension to Multiple Marginals:** Multi-Marginal Optimal Transport (MMOT) is a natural extension of OT aiming to find a joint distribution $\pi \in \Pi(\rho_0, \rho_1, \ldots, \rho_M)$, where $\Pi$ is the set of all joint distributions with marginals $\{\rho_i\}_{i=0}^{M}$, minimizing $W_2^2(\rho_i, \rho_j)$ for all $i \neq j$ (Pass, 2015). Of these $\pi$, we are interested in the special case where there exists a total order on the time labels $t_i$ associated with each $\rho_i$. We henceforth use MMOT to refer to this ordered MMOT case. Instances referring to the fully joint MMOT will be explicitly referred to as such.

Extending our least action principle to the MMOT case with arbitrary time points $t_0, t_1, \ldots, t_M$, we pose the same question: among all possible stochastic processes that could connect the observed marginal distributions $\{\rho_i\}_{i=0}^{M}$, which one is the most probable given our prior knowledge? This leads us to formulate the problem as finding the drift $u_t(x)$ that minimizes the cumulative transport cost and provides the smoothest and most efficient flow connecting the observed distributions over time, while ensuring robustness against overfitting. In summary, we require:

- Scalability in High Dimensions: While directly solving high-dimensional transport problems is computationally challenging (Benamou & Brenier, 2000; Peyré & Cuturi, 2019), our approach efficiently approximates the solution. By leveraging advances in simulation-free score and flow matching methods, we model the high-dimensional stochastic process directly in the ambient space, avoiding dimensionality reduction strategies that could obscure important dynamical features.

- Robustness Against Overfitting: By minimizing the total transport cost across all time intervals, we introduce only essential adjustments to match the observed marginals, preventing the model from overfitting to limited observations and ensuring that the inferred dynamics generalize well beyond the training data.

- Insensitivity to the Timing of Snapshots: The formulation inherently accommodates arbitrary and irregular time points $t_i$, making it robust to the choice of measurement times. By focusing on the minimal action path that passes through the observed marginals, we capture the system's evolution without being constrained by the timing of data collection.

### 1.2. Literature Review

Direct learning of the high-dimensional partial differential equation (2) is computationally prohibitive due to the complexity of integration and divergence computations in high-dimensional spaces (Benamou & Brenier, 2000; Peyré & Cuturi, 2019). Hence, current approaches typically consider reduced-dimensional data representations with gradient-based drifts originating from developmental biology (Weinreb et al., 2018; Schiebinger et al., 2019) where the focus is primarily on slow time scales and the assumption of low-dimensional manifold dynamics is often useful. In this context, dimensionality reduction tools such as t-SNE (Van der Maaten & Hinton, 2008), UMAP (McInnes et al., 2018), and PHATE (Moon et al., 2019) are extensively used to simplify the modeling. However, these techniques can obscure critical faster-scale dynamical information, introduce artifacts (Kiselev et al., 2019), and result in the loss of important biological information in the reduced, folded space.

Neural Ordinary Differential Equations (Neural ODEs) have emerged as a powerful tool for modeling continuous-time dynamics and connecting probability measures over time (Chen et al., 2018a). This approach offers an alternative method by parameterizing the time derivative of the hidden

state with a neural network, which is trained to approximate the drift term in the Fokker-Planck equation (2). While this method has been successfully applied (Tong et al., 2020; Huguet et al., 2022) for modeling cellular dynamics and trajectory inference, it still operates primarily in reduced-dimensional spaces.

Recent multi-marginal approaches have attempted to handle multiple time points simultaneously. Chen et al. (2024) developed a deep multi-marginal momentum Schrödinger bridge approach that, while capable of working in high dimensions, requires expensive flow integration and memory-intensive caching of trajectories during training. Similarly, Albergo et al. (2024) proposed stochastic interpolants for multi-marginal modeling but still relies on ODE/SDE integration and marginal distributions as supervision signals, which becomes computationally challenging in high dimensions. These approaches share common limitations: they either require dimension reduction to handle computational complexity, or they depend on expensive numerical integration and trajectory generation during training.

Alternative approaches using generative models attempt to transform a simple distribution to an arbitrary target distribution. Variational Autoencoders (VAEs) (Kingma et al., 2013) learn an encoder-decoder pair, $q(z \mid x)$ and $p(x \mid z)$, such that the decoder can generate $x \sim \rho_1$ given samples $z \sim \rho_0$. Generative Adversarial Networks (GANs) (Goodfellow et al., 2014) employ a generator-discriminator framework, where the generator $G(z)$ produces $x = G(z)$ with $x \sim \rho_1$ for $z \sim \rho_0$. While successful, these methods are limited by the simplicity of the source distribution $\rho_0$, often chosen to be uniform or normal for analytical convenience. Moreover, these models represent static transformations with no notion of time and cannot generate intermediate states at arbitrary time points, making them unsuitable for modeling dynamic processes where temporal evolution is crucial.

Although diffusion models (Ho et al., 2020; Song & Ermon, 2019) incorporate a time component by learning a denoising Markovian reverse process, their notion of "time" corresponds to a noise schedule rather than physical time. This limitation prevents them from capturing actual temporal dynamics or generating data at arbitrary time points not specified during training.

**Our Approach** We introduce Multi-Marginal Stochastic Flow Matching (MMSFM) to address these limitations by adapting recent developments in simulation-free approaches (Lipman et al., 2023; Tong et al., 2024b) to our setting. These methods learn $p_t$ directly in the ambient space without dimensionality reduction or explicit simulation. However, their direct application to our multi-marginal setting requires careful adaptation to principles described in

Section 1.1 to ensure robust learning and prevent overfitting.

Our key innovation lies in learning continuous spline measures through overlapping windows of consecutive marginals during training. Specifically, we process overlapping triplets $(\rho_i, \rho_{i+1}, \rho_{i+2})$ in a rolling fashion, where we demonstrate in Section 2.3 that a window size of two strikes an optimal balance between enforcing smoothness constraints and computational efficiency. This approach enables us to capture local dynamics across uneven time intervals, maintain consistency between overlapping windows, and generate intermediate states between observed snapshots, effectively creating a "motion picture" of the system's evolution. The overlapping nature of these learned flows ensures robustness against the specific choice of measurement times while preserving the high-dimensional structure of the data.

Although we were unaware during the development of our method, an important concurrent work by Rohbeck et al. (2025) also considers a very similar approach to ours. We include a more in-depth discussion regarding similarities and differences between our approaches in Section 2.3.3.

## 2. Problem Formulation and Methodology

Formally, let $0 = t_0 < t_1 < \cdots < t_M = 1$ denote a sequence of normalized time points in [0, 1], and let $\rho_i$ be the continuous probability distribution of the system state at time $t_i$ in Euclidean space $\mathbb{R}^d$. Our data consists of snapshot samples $X_{t_i} = \{x^{(j)} : x \sim \rho_i\}_{j=1}^{N_i}$, at these time points. The goal is to learn a continuous probability path $p_t(x)$ for $t \in [0, 1]$ satisfying $p_{t_i} = \rho_i$ for all $i$, describing the evolution of the system over time.

### 2.1. Dynamic formulation of the Wasserstein distance and Wasserstein Splines

Benamou & Brenier (2000) introduced a dynamic formulation of the Wasserstein distance, connecting OT with fluid dynamics:

$$W_2^2(\mu, \nu) = \inf_{\substack{p_t, u_t \\ \frac{\partial p_t}{\partial t} + \nabla \cdot (p_t u_t) = 0 \\ p_0 = \mu, \ p_1 = \nu}} \int_0^1 \int_{\mathbb{R}^d} \|u_t(x)\|^2 \, p_t(x) \, dx \, dt$$

While the fully joint MMOT extends this framework to multiple distributions, computing the MMOT plans becomes computationally challenging in high dimensions. Prior work (Chen et al., 2018b; Benamou et al., 2019) examined the formulation

$$\inf_{X_t} \int_0^1 \mathbb{E}\left[\left\|\ddot{X}_t\right\|^2\right] dt, \tag{5}$$

termed P-splines by Chewi et al. (2021). Unfortunately, this does not fit our needs because $X_t$ here is considered to be

a stochastic process whereas we need a deterministic flow. Moreover, these formulations are still quite computationally expensive given that we need to solve this problem within the training loop. Instead, Chewi et al. (2021) proposed *transport splines* as a method to efficiently obtain deterministic maps that smoothly interpolate between multiple distributions. The key idea is to sample points from the distributions $\rho_i$ and apply a Euclidean interpolation algorithm between these points. The specific spline algorithm is left as a design choice for the user. Options include the natural cubic spline interpolation which minimizes the integral of the squared acceleration

$$\inf_{\gamma_t} \int_0^1 \mathbb{E}\left[\|\ddot{\gamma}_t\|^2\right] dt, \tag{6}$$

where $\gamma_t$ denotes a curve in space, and the cubic Hermite spline (Hermite & Borchardt, 1878) which represents each interval $(x_i, x_{i+1})$ as the third-degree polynomial

$$\begin{aligned} X(t) =&(2t^3 - 3t^2 + 1)x_i + (t^3 - 2t^2 + t)x_i' \\ &+ (-2t^3 + 3t^2)x_{i+1} + (t^3 - t^2)x_{i+1}' \end{aligned} \tag{7}$$

where $x_0$, $x_1$ are the boundary constraints, and $x_0'$, $x_1'$ are the derivatives w.r.t. time at those points.

In practice, we adopt transport splines using compositions of probabilistic OT plans in place of deterministic OT maps. Let the joint distribution $\pi$ be the MMOT plan over $(x_{t_0}, x_{t_1}, \ldots, x_{t_M})$. Although the true MMOT plan involves all pairs of marginals, we are interested in the case where there is a temporal ordering. This structure allows us to take advantage of a first-order Markov approximation and decompose $\pi$ into conditional plans

$$\pi(x_{t_0}, \ldots, x_{t_M}) \approx \pi(x_{t_0}, x_{t_1}) \prod_{i=2}^{M} \pi(x_{t_i} \mid x_{t_{i-1}}), \tag{8}$$

where $\pi(x_{t_i} \mid x_{t_{i-1}})$ specifies how to transport a point $x_{t_{i-1}} \sim \rho_{i-1}$ to the distribution $\rho_i$. We obtain the conditional plan as $\pi(x \mid y) = \pi(x, y)/p(y)$ where $p(y) = \int \pi(x, y) dx$ is the marginal distribution of $y$. By applying the transport spline procedure to batches of vectors $(X_{t_i})_{i=0}^{M}$, the conditional plans act as alignment operators, allowing us to construct Euclidean splines through optimally coupled points $(X_{t_i}^\star)_{i=0}^{M}$.

## 2.2. Simulation-Free Score and Flow Matching

We aim to model the stochastic process bridging the multiple distributions $\rho_i$ by learning the underlying dynamics of the system in Equation (1) and the associated Fokker-Planck equation (2). Tong et al. (2024b) introduced a reparameterization of the drift $u_t(x)$ as

$$u_t(x) = u_t^\circ(x) + \frac{g^2(t)}{2} \nabla \log p_t(x), \tag{9}$$

where $u_t^\circ(x)$ is the deterministic component, and $\nabla \log p_t(x)$ is the score function of the density $p_t(x)$. This observation allows us to decouple the learning of the deterministic drift $u_t^\circ(x)$ and the score function $\nabla \log p_t(x)$. Therefore, specifying $u_t^\circ(x)$ and $\nabla \log p_t(x)$ is sufficient to define the SDE drift $u_t(x)$. Tong et al. (2024b) proposed the unconditional score and flow matching objective:

$$\begin{aligned} \mathcal{L}(\theta) =&\mathbb{E}_{t\sim\mathcal{U}(0,1), x\sim p_t(x)} \Big[ \|v_t(x; \theta) - u_t^\circ(x)\|^2 + \\ &\lambda(t)^2 \|s_t(x; \theta) - \nabla \log p_t(x)\|^2 \Big], \end{aligned} \tag{10}$$

where $v_t(x; \theta)$ and $s_t(x; \theta)$ are neural networks approximating the drift and score functions, respectively, and $\lambda(t)$ is a weighting function. However, $p_t(x)$ is unknown and thus directly computing $u_t^\circ(x)$ and $\nabla \log p_t(x)$ is challenging. To overcome this, Tong et al. (2024b) proposed a conditional formulation of the loss function:

$$\begin{aligned} \mathcal{L}(\theta) =&\mathbb{E}_{t\sim\mathcal{U}(0,1), z\sim q(z), x\sim p_t(x|z)} \Big[ \|v_t(x; \theta) - u_t^\circ(x|z)\|^2 \\ &+\lambda(t)^2 \|s_t(x; \theta) - \nabla \log p_t(x|z)\|^2 \Big], \end{aligned} \tag{11}$$

where $z$ represents conditioning variables, and $x \sim p_t(x|z)$. In this conditional framework, $u_t^\circ(x|z)$ and $\nabla \log p_t(x|z)$ can be computed analytically or estimated empirically based on the conditional distribution $p_t(x|z)$. We can reconstruct the learned SDE drift using:

$$u_t(x; \theta) = v_t(x; \theta) + \frac{g^2(t)}{2} s_t(x; \theta), \tag{12}$$

and integrate it with given initial conditions $x_0$ to infer the trajectories that develop from those initial conditions.

## 2.3. Learning Overlapping Mini-Flows for Multi-Marginal Data

We aim to train an ODE drift network $v_t(x; \theta)$ and a score network $s_t(x; \theta)$ to learn an overall flow based on the *mini-flows* on overlapping $(k + 1)$-tuples $(\rho_i, \rho_{i+1}, \ldots, \rho_{i+k})$ for $i = 0, 1, \ldots, M - k$ in a rolling window fashion. Because transport splines are ultimately just approximations for the true MMOT, the rolling windows provide a variation of perturbations in the approximated error from any single geodesic spline segment estimate. See Figure 1 for a visual representation of the variation of paths in an interval.

**Theorem 2.1.** *The gradient of the loss for a single interval $(t_i, t_{i+1})$ with $k$ overlapping mini-flows is given by*

$$\begin{aligned} \nabla_\theta \mathcal{L}_{t_i:t_{i+1}} =&\nabla_\theta \mathbb{E} \Bigg[ \Bigg\| v_t(x; \theta) - \sum_{j=0}^{k-1} u_t(x \mid z_{t_{i-j}:t_{i-j+k}}) \Bigg\|^2 \\ &+ (k-1)\|v_t(x; \theta)\|^2 \Bigg] \end{aligned}$$

*where the expectation is taken over* $q(z_{t_{i-k+1}:t_{i+k}})$, $\mathcal{U}(t_i, t_{i+1})$, *and* $\tilde{p}(x \mid z_{t_{i-k+1}:t_{i+k}}) = \frac{1}{k} \sum_{j=0}^{k-1} p_t(x \mid z_{t_{i-j}:t_{i-j+k}})$.

Geometrically, the neural network learns the direction of the overall movement given by the sum of the $u_t$ vectors. The magnitude of this movement is scaled down by the competing objective $(k-1)\|v_t(x; \theta)\|^2$.

**Corollary 2.2.** *If all the mini-flow regression signals* $u_t(x \mid z_{t_{i-j}:t_{i-j+k}})$ *are equal, then the single interval loss recovers the CFM loss (Theorem 2.1 of* Lipman et al. (2023)*) up to a scalar factor.*

We provide proofs in Appendix A.

Our method handles overlapping trajectories where $u_{t_i}^{\circ}(x) \neq u_{t_j}^{\circ}(x)$ for fixed $x$ and $t_i \neq t_j$, accommodating the possibility that trajectories may cross over a point at different times in multi-marginal settings. In practice, we train using mini-batches of size $b$.

By incorporating stochasticity through score matching, we improve robustness and avoid overfitting in high-dimensional spaces. The score $\nabla_x \log p_t(x|z)$ allows the model to capture the inherent uncertainty and variability in the data. Using the identity $\nabla_x \log p_t(x|z) = \nabla_x p_t(x|z)/p_t(x|z)$, we see that the score nudges predictions towards more likely regions, thereby implicitly exploring the region around the local per-sample flow. This efficiency allows us to remain in the ambient dimension $d$ and sidestep dimensionality reduction strategies which often introduce information loss and additional complexities into the flow dynamics. Moreover, re-projecting the trajectories back into the ambient space introduces undesirable reconstruction artifacts.

### 2.3.1. TRANSPORT SPLINES SAMPLING OF $z$ AND STRATIFIED SAMPLING OF $t$

We sample $z$ from a MMOT plan $\pi$ using transport splines by first drawing samples $X_{t_i}, X_{t_{i+1}}, \ldots, X_{t_{i+k}} \sim \rho_i, \rho_{i+1}, \ldots, \rho_{i+k}$, where each $X_{t_i}$ is a batch of i.i.d. samples from $\rho_i$. Then, we compute the MMOT plan given by the first-order Markov approximation (8):

$$\pi(x_{t_i}, \ldots, x_{t_{i+k}}) \approx \pi(x_{t_i}, x_{t_{i+1}}) \prod_{j=i+2}^{i+k} \pi(x_{t_j} \mid x_{t_{j-1}}).$$

The initial plan $\pi(x_{t_i}, x_{t_{i+1}})$ is a standard OT plan w.r.t. the squared Euclidean distance $\|x_{t_i} - x_{t_{i+1}}\|^2$ as the cost function. Next, we compute the conditional map $\pi(x_{t_j} \mid x_{t_{j-1}})$ using

$$\pi(x_{t_j} \mid x_{t_{j-1}}) = \frac{\pi(x_{t_{j-1}}, x_{t_j})}{\pi(x_{t_{j-1}})} = \frac{\pi(x_{t_{j-1}}, x_{t_j})}{\int \pi(x_{t_{j-1}}, x_{t_j}) \, dx_{t_j}}.$$

We refer the reader to Appendix C.1 for implementation details.

In the original source-target distribution pair setting, we sample $t \sim \mathcal{U}(0, 1)$. To accommodate our mini-flow method, we could sample $t \sim \mathcal{U}(t_i, t_{i+k})$ for the $i$th mini-flow. However, this approach is ineffective for training uneven time intervals—for example $t_{i+1} - t_i \ll t_{i+2} - t_{i+1}$—leading to insufficient sampling from the smaller interval. To handle this, we adopt a stratified sampling strategy, sampling an equal number of time points from $\mathcal{U}(t_i, t_{i+1}), \mathcal{U}(t_{i+1}, t_{i+2})$, and so on to ensure balanced training across intervals. Specifically, for a total batch size of $b$, we sample $b/k$ time points from each interval.

### 2.3.2. MINI-FLOW ODE AND SCORE REGRESSION TARGETS

Theorem 3 of Lipman et al. (2023) and Theorem 2.1 of Tong et al. (2024a) derive the ODE flow regression target for a conditional Gaussian probability path $p_t(x \mid z) = \mathcal{N}(x \mid \mu_t, \sigma_t^2)$ as

$$u_t^{\circ}(x \mid z) = \frac{\sigma_t'}{\sigma_t}(x - \mu_t) + \mu_t' \tag{13}$$

where $\mu_t$ and $\sigma_t$ are respectively the time-varying mean and standard deviation of the flow conditioned on $z$. The prime notation ($'$) denotes differentiation w.r.t. time $t$. We set $\mu_t = \mu_{i:i+k}(t)$ for a transport spline $\mu_{i:i+k}(t) : [t_i, t_{i+k}] \to \mathbb{R}^d$, constructed through the points in $z$ via Euclidean spline interpolation.

For $\sigma_t$ we consider the case of Brownian bridges with constant diffusion $g(t) = \sigma$ and set $\sigma_t = \sigma\sqrt{t(1-t)}$ along the global time $t \in [0, 1]$. Alternatively, we can set $\sigma_t$ based on the Brownian bridge of the mini-flow from $a = t_i$ to $b = t_{i+k}$, reparameterizing as $\sigma_t = \sigma\sqrt{r(t)(1 - r(t))}$, where $r(t) = \frac{t-a}{b-a}$. In this case, the derivative $\sigma_t'$ must take into account the reparameterization, yielding $\sigma_t' = \frac{d\sigma_t}{dr} \cdot \frac{dr}{dt} = \frac{d\sigma_t}{dr} \cdot \frac{1}{b-a}$. Because $\mu_t, \sigma_t, \mu_t', \sigma_t'$ can be expressed analytically, we can directly compute these quantities and efficiently compute the regression target $u_t^{\circ}$ using Equation (13). Given our Gaussian probability path, we can easily derive the score regression target as $\nabla \log p_t(x \mid z) = \frac{\mu_t - x}{\sigma_t^2}$, or alternatively $-\frac{\epsilon}{\sigma_t}$ for $\epsilon \sim \mathcal{N}(0, I)$.

We summarize our method in Algorithm 1. Once we have the trained networks $v_t(x; \theta), s_t(x; \theta)$, we can construct the SDE drift $u_t(x; \theta)$ using (12), and generate trajectories from given initial conditions $x_0$ by an SDE integration with drift $u_t(x; \theta)$ and diffusion $\sigma$.

### 2.3.3. WINDOW SIZE $k$ AND SPLINE ALGORITHM

We choose our window size $k = 2$ based on the properties of our chosen Euclidean spline algorithm and considerations

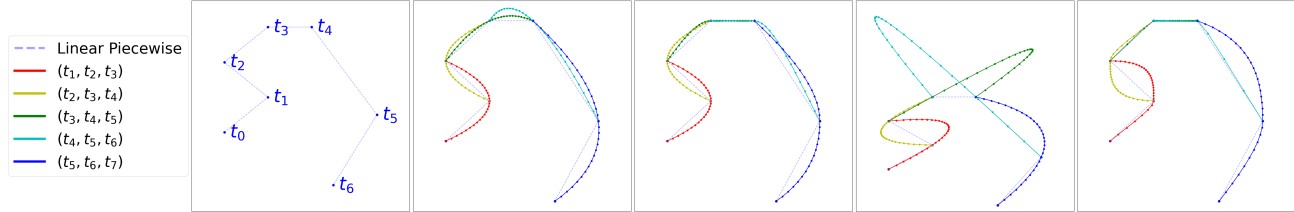

*Figure 1.* Comparison of Euclidean splines on overlapping windows of size $k = 2$, demonstrating the potential for overlapping windows to capture variations of paths though the same intervals. From left to right: 1) The 7 points to interpolate with time labels $t_0, \ldots, t_6$. 2) Natural cubic splines on equidistant time intervals. 3) Monotonic cubic Hermite splines on equidistant time intervals. 4) Natural cubic splines on arbitrary time intervals $\mathcal{T} = (0, 0.05, 0.2, 0.27, 0.86, 0.95, 1)$. Note the overshooting required to satisfy the continuity of $S''$ at $t_3$ and $t_4$. 5) Monotonic cubic Hermite splines on arbitrary time intervals.

---

**Algorithm 1** MMSFM Training

---

> **Input:** Training data, $k, \sigma$
> Initialize networks $v_t(x; \theta), s_t(x; \theta)$
> Set $g(t) \leftarrow \sigma$
> Set $\sigma_t \leftarrow \sigma\sqrt{t(1-t)}$ or $\sigma\sqrt{r(t)(1-r(t))}$
> Set $\lambda(t) \leftarrow \dfrac{2\sigma_t}{g(t)^2}$
> **while** training **do**
>     **for** $i = 0$ **to** $M - k$ **do**
>         Sample mini-batches:
>         $X_{t_i}, \ldots, X_{t_{i+k}} \sim \rho_i, \ldots, \rho_{i+k}$
>         Compute OT plans:
>         $\pi(X_{t_i}, X_{t_{i+1}}), \pi(X_{t_{i+2}} \mid X_{t_{i+1}}), \ldots$
>         Generate $z \leftarrow (X^\star_{t_i}, \ldots, X^\star_{t_{i+k}})$ using $\pi$
>         Compute $\mu_t \leftarrow \mu_{i:i+k}(t)$
>         Set $p_t(x \mid z) \leftarrow \mathcal{N}(x \mid \mu_t, \sigma_t^2)$
>         Sample times $t$ from $[t_i, t_{i+k}]$
>         Sample $x \sim p_t(x \mid z)$
>         Compute $u_t^\circ(x \mid z) \leftarrow \dfrac{\sigma_t'}{\sigma_t}(x - \mu_t) + \mu_t'$
>         Compute $\nabla \log p_t(x \mid z) \leftarrow \dfrac{\mu_t - x}{\sigma_t^2}$
>         Compute $\mathcal{L}(\theta)$ from (11)
>         Update $\theta$ using $\mathcal{L}(\theta)$
>     **end for**
> **end while**
> **Output:** Trained networks $v_t(x; \theta), s_t(x; \theta)$

---

to the running time. We opt to use monotonic cubic Hermite splines instead of natural cubic splines for four main reasons. First, the guaranteed monotonicity of each piecewise cubic polynomial ensures no overshoot within each dimension, thus removing overshooting when computing the ODE flow regression target (13). Second, monotonic cubic Hermite splines by construction do not necessarily have a continuous second derivative. While this is a desired property for smoother curves (and in fact enforced for natural cubic splines), this condition can restrict the curve from taking a

more direct path such as from a linear piecewise interpolation. Third, while using a larger window can potentially fit a spline closer to the linear piecewise interpolation, allowing for smaller windows can better capture a wider variation of paths, increasing the robustness of the learned flow. Moreover, 3 control points are sufficient to learn a curvature at the interior control point. Fourth, the specific coefficients describing each piecewise cubic polynomial are efficient to compute, scaling linearly in $\mathcal{O}(k)$ with the $k + 1$ points to interpolate. For a window size $k$ and $M$ time points, the overlapping window routine computes $(M - k)k$ splines resulting in a total complexity of $\mathcal{O}((M - k)k)$. This is "maximized" when $k = M/2$ for a complexity of $\mathcal{O}(M^2)$, and "minimized" when $k = 1$ or $k = M - 1$ for a complexity of $\mathcal{O}(M)$. As an added bonus, monotonic cubic Hermite splines are highly insensitive to control points that are not immediate neighbors. Thus, choosing a larger window size $k > 2$ does not meaningfully increase the amount of information captured by the spline.

Rohbeck et al. (2025) considers a similar approach arriving at the same ordered MMOT plan approximation, detailed in their Appendix B. Moreover, Rohbeck et al. also use splines as the interpolation method allowing for irregular time points, but differ in their selection of natural cubic splines over all the time points ($k = M - 1$). This contrasts with our choice to use monotonic cubic Hermite splines on rolling windows over $k = 2$. Although natural cubic splines may be more analytically tractable, we reiterate that the monotonic cubic Hermite spline nonetheless provides a monotonicity guarantee within each dimension and thus avoids overshoot. This property is especially relevant when exploring the sensitivity of our method to highly irregular time points. We attribute the overshooting behavior in natural cubic splines to the presence of neighboring intervals with vastly different lengths. For example, consider the time point sequence $0.2 \rightarrow 0.27 \rightarrow 0.86$ in Figure 1. The two corresponding intervals have length $0.07$ and $0.59$, nearly a 9-fold difference. For the natural cubic spline, any change in

velocity and acceleration along the short interval can happen relatively quickly, but the corresponding change for the long interval must be drawn out. In particular, the continuity of the acceleration constraint does not help in this regard as it prevents the spline from instantaneously re-adjusting velocities as necessary. We include a more in-depth discussion of splines in Appendix B.1.

## 3. Results

We briefly describe our data and setup below, and also include a more detailed experimental setup description in Appendix D. We summarize our results in Tables 1 and 2.

### 3.1. Experimental Setup

We applied our rolling window framework to three synthetic datasets, two RNA gene expression datasets, and an image classification dataset. From our framework we use the $k = 1$ (Pairwise, equivalent to SF2M (Tong et al., 2024b)) and $k = 2$ (Triplet) mini-flow settings. We approximate the MMOT plan with transport splines computed on mini-batch OT given the smaller computation cost and asymptotic convergence properties (Fatras et al., 2020; 2021). We additionally use MIOFlow (Huguet et al., 2022) on the synthetic datasets to examine the difference in performance for ambient-space and latent-space models. The initial conditions for the generated trajectories are from a held-out set of samples from the source distribution $\rho_0$. Evaluations on all synthetic and RNA gene expression datasets are computed by leaving out a time point marginal during training and calculating the Wasserstein metrics $W_1$ and $W_2^2$ (using Euclidean distance as the cost function), the maximum mean discrepancy with a mixture kernel (MMD(M)), and the maximum mean discrepancy using a Gaussian kernel (MMD(G)) at the left-out time point. For the image dataset, we train using all time point marginals and examine the training stability and loss over epochs.

**Synthetic Data:** Our three synthetic datasets are the S-shaped Gaussians, the $\alpha$-shaped Gaussians, and a synthetic scRNA dataset generated by the DynGen simulator (Cannoodt et al., 2021) which we repurpose from Huguet et al. (2022). The S and $\alpha$-shaped Gaussians both consist of 7 marginal distributions in $\mathbb{R}^2$. We select these three datasets because S-shaped Gaussians involve learning a flow with changing curvature, the $\alpha$-shaped Gaussians have a cross-over point for some $x$ where the flow $u_{t_i}(x) \neq u_{t_j}(x)$ and $i \neq j$, and the DynGen dataset introduces a bifurcation. We evaluate both Gaussian datasets on three different time point labels: equidistant time points $\mathcal{T}_1 = (0, 0.17, 0.33, 0.5, 0.67, 0.83, 1)$, a first set of arbitrary time points $\mathcal{T}_2 = (0, 0.08, 0.38, 0.42, 0.54, 0.85, 1)$, and a second set of arbitrary time points $\mathcal{T}_3 =$

*Table 1.* Comparison of the inferred distributions generated by MIOFlow and our method using Pairwise and Triplet mini-flows at the held-out time point. For the equidistant time points $\mathcal{T}_1$, we hold out $t_5 = 0.83$ and $t_4 = 0.67$ respectively for the S-shaped and $\alpha$-shaped data. We do the same for the arbitrary time points $\mathcal{T}_2$, holding out $t_5 = 0.85$ and $t_4 = 0.54$. We also examine distance metrics averaged across all time points for $\mathcal{T}_3$. From DynGen, we hold out $t_1 = 0.25$.

|  |  | S-SHAPE (HOLD $t_5$) | | | $\alpha$-SHAPE (HOLD $t_4$) | | |
|  |  | MIOFLOW | PAIR | TRIP | MIOFLOW | PAIR | TRIP |
|---|---|---|---|---|---|---|---|
| $\mathcal{T}_1$ | $W_1$ | 8.16 | 2.36 | **1.83** | 21.54 | **3.78** | 4.54 |
|  | $W_2^2$ | 66.91 | 5.87 | **3.86** | 464.36 | **14.56** | 21.06 |
|  | MMD(G) | 7.26 | 2.29 | **1.47** | 7.65 | **3.96** | 4.26 |
|  | MMD(M) | 66.19 | 5.24 | **3.11** | 463.66 | **13.92** | 20.01 |
| $\mathcal{T}_2$ | $W_1$ | 9.42 | 2.12 | **1.62** | 5.04 | 8.08 | **3.79** |
|  | $W_2^2$ | 89.07 | 4.56 | **2.73** | 25.85 | 76.82 | **14.73** |
|  | MMD(G) | 7.37 | 2.36 | **1.53** | 6.46 | 4.01 | **3.77** |
|  | MMD(M) | 88.37 | 4.12 | **2.22** | 25.35 | 64.81 | **14.07** |
|  |  | S-SHAPE (ALL) | | | $\alpha$-SHAPE (ALL) | | |
| $\mathcal{T}_3$ | $W_1$ | — | 12.06 | **3.71** | — | **33.95** | 186.68 |
|  | $W_2^2$ | — | 257.86 | **35.01** | — | **2400.15** | 2.51E6 |
|  | MMD(G) | — | 4.12 | **2.12** | — | 4.62 | **1.35** |
|  | MMD(M) | — | 241.01 | **7.87** | — | **2168.01** | 7.76E5 |
|  |  | DYNGEN (HOLD $t_1$) | | | — | | |
|  | $W_1$ | 0.85 | **0.74** | 0.83 |  |  |  |
|  | $W_2^2$ | 0.98 | **0.63** | 0.82 |  |  |  |
|  | MMD(G) | 0.53 | 0.38 | **0.22** |  |  |  |
|  | MMD(M) | 0.51 | 0.19 | **0.10** |  |  |  |

$(0, 0.2, 0.27, 0.3, 0.88, 0.98, 1)$ with neighboring small and large intervals. We introduce $\mathcal{T}_3$ as a highly irregular time point set with a very small interval of 0.03 followed by a disproportionately large interval of 0.58 in $0.27 \rightarrow 0.3 \rightarrow 0.88$. Also included is a second miniscule gap of 0.02 in the last interval $0.98 \rightarrow 1$. The DynGen dataset has 5 marginal distributions on equidistant time points $\mathcal{T} = (0, 0.25, 0.5, 0.75, 1)$.

**Real Data:** We consider gene expression data from the Multiome and CITEseq datasets published as part of a NeurIPS competition (Burkhardt et al., 2022). Measurements are taken at $\mathcal{T} = (2, 3, 4, 7)$ days, which again normalize to $\mathcal{T} = (0, 0.2, 0.4, 1)$. We follow the procedure in (Tong et al., 2024b) and preprocess the data into the first 50 and 100 principal components, along with the top 1000 highly variable genes (Satija et al., 2015; Stuart et al., 2019; Zheng et al., 2017).

Finally, we consider a generative perspective and look at the performance of our model on image progression through various classes from the Imagenette dataset (Jeremy Howard), a subset of 10 easily classified classes from the ImageNet dataset (Deng et al., 2009). Specifically, we look at the progression: gas pump to golf ball to parachute. For the Pairwise model we set $\mathcal{T} = (0, 0.25, 1)$, and for the Triplet model we set $\mathcal{T} = (0, 0.5, 1)$.

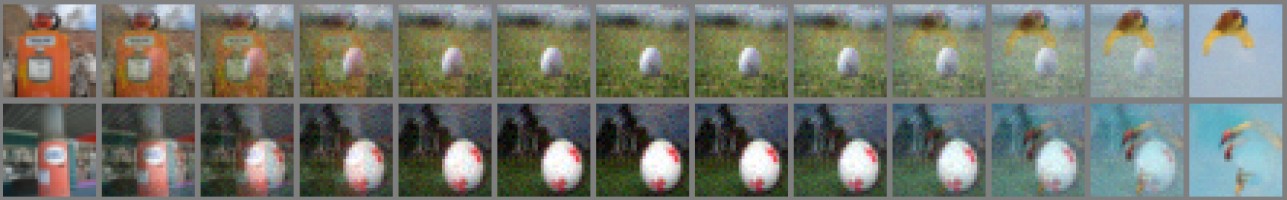

*Figure 2.* Example trajectories for a $32 \times 32$ pixel image progression through the Imagenette classes (gas pump $\rightarrow$ golf ball $\rightarrow$ parachute). Results are generated using our Triplet model with an equidistant time scheme.

*Table 2.* Comparison of the Pairwise and Triplet methods on the CITEseq and Multiome gene expression datasets. We hold out $t_2 = 0.4$ for both datasets.

|  |  | PCA 50 | | PCA 100 | | HI-VAR 1000 | |
|---|---|---|---|---|---|---|---|
|  |  | PAIR | TRIP | PAIR | TRIP | PAIR | TRIP |
| CITESEQ | $W_1$ | 54.18 | **53.98** | 62.85 | **62.08** | **50.64** | 50.71 |
|  | $W_2^2$ | 3027.28 | **3019.89** | 4036.41 | **3942.08** | **2579.84** | 2585.98 |
|  | MMD(G) | 0.16 | 0.16 | 0.16 | **0.15** | 0.05 | 0.05 |
|  | MMD(M) | 339.20 | **344.89** | 345.09 | **331.72** | **48.53** | 49.83 |
| MULTI | $W_1$ | 61.79 | **60.92** | 70.72 | **70.39** | 56.15 | **56.10** |
|  | $W_2^2$ | 3918.50 | **3806.89** | 5077.07 | **5029.56** | 3166.01 | **3160.84** |
|  | MMD(G) | 0.30 | **0.27** | 0.25 | **0.23** | 0.04 | 0.04 |
|  | MMD(M) | 793.34 | **705.21** | 656.86 | **621.32** | 40.71 | **40.29** |

### 3.2. Discussion

Learned flows are visualized in Appendix F. Our method consistently outperformed MIOFlow on the interpolation at the held-out time point for the synthetic data. Interestingly, the Pairwise model slightly outperformed the Triplet model for the $\alpha$-shaped Gaussians on $\mathcal{T}_1$. We believe that in this specific instance, the masked time point corresponded to an interval where the momentum from the prior interval was enough for the Pairwise model to infer the held-out marginal. In contrast, the $\alpha$-shaped Gaussians on $\mathcal{T}_2$ show the Triplet model outperformed the Pairwise model by a significant margin; even MIOFlow generally outperformed the Pairwise model in this instance. This suggests that the Triplet method is more effective for non-equidistant time snapshots especially when capturing complex temporal dynamics because the variation of flows provided by splines in overlapping windows helps learn the held-out marginal. The success of our methods on $\mathcal{T}_2$ demonstrates the robustness and stability of our approach even when handling arbitrary time points. Looking at the trajectory plots, we can also confirm that our method is able to handle datasets with varying flow curvatures and flow cross-overs.

The bifurcating flow of DynGen posed a challenge for our models: while they outperformed MIOFlow on the metrics, but struggled to handle the bifurcating trajectories. We suspect this behavior to stem from mini-batch OT because it does not enforce a consistency constraint on the sampling process, resulting in cases where particles are able to jump

between separate branches of the bifurcated flow. We did not explore methods to mitigate this problem and believe this to be an avenue for future work.

Remaining in the high-dimensional ambient space without aggressive dimensionality reduction preserves important biological information, leading to more biologically plausible trajectories. The ability to generate samples at arbitrary time points allows us to explore the system's behavior beyond the observed data, potentially identifying critical time windows where intervention might be most effective. This has implications for understanding drug resistance mechanisms and designing more effective therapeutic strategies.

In all instances, MIOFlow generated idiosyncratic trajectories which matched the marginals at the specified time points but performed poorly between those time points. We believe this to be the case because MIOFlow operates in the embedding space generated by a GAE. This structure works very well for trajectories in the embedding space but poses a problem when reconstructing the trajectories in the ambient space. The GAE is only trained on the data marginals $\{\rho_i\}_{i=0}^M$ at times $\{t_i\}_{i=0}^M$, which means that data points not specified in the data are effectively out-of-distribution w.r.t. the GAE. These out-of-distribution points arise naturally from generating trajectories which spend time traveling between the data distributions. In addition, we notice that all the reconstructed points exhibit high bias and low variance, tending to be bunched very close to each other. This quality perhaps captures the first moment well but not any higher moments.

We validate our Triplet model's ability to learn flows on high dimensional, noisy data, taken at irregular time points by comparing results from the Pairwise and Triplet models on the CITEseq and Multiome datasets. These two datasets contain high dimensional samples with noise inherent to biological measurements and measurements from non-uniform time intervals. We see the Triplet model successfully outperform the Pairwise model on inferring the distribution at the held out time point even in these conditions.

On Imagenette, we examine our Triplet model's performance against that of the Pairwise model both by the train-

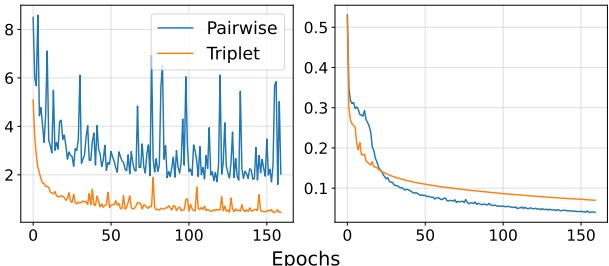

*Figure 3.* Loss plots for the Triplet and Pairwise models on the Imagenette dataset. The x-axis represents the epoch (1000 gradient steps), and the y-axis represents the mean loss value for that epoch. Left) Flow loss. Right) Score loss.

ing loss curves and visually. In both cases, we used the model state after training for 160k gradient steps. The Triplet model performed better than the Pairwise model with respect to the flow loss and fairly even on the score matching loss. Both models were able to generate image trajectories which pass through the golf ball distribution, but the Pairwise model exhibited more difficulty than the Triplet model in going to the parachute distribution. This suggests the Triplet model exhibited more stable training, further validating our approach.

Finally, we examine performance of the Pairwise and Triplet models on the S and $\alpha$-shaped datasets using the highly unbalanced time points $\mathcal{T}_3$. Here, we find that the Triplet model greatly outperforms the Pairwise model on the overall learned flow for the S-shaped dataset, but seemingly vastly underperforms for the $\alpha$-shaped dataset. By taking a closer look at the visualizations of the learned flow, we can see that in both cases, the short-long-short interval pattern poses a significant difficulty, suggesting that arbitrary time points do indeed increase the difficulty of the learning task. In the S-shaped case, the Pairwise model completely fails to learn this trajectory, whereas the Triplet model does better, learning to speed up and then slow down between $t_2 = 0.27$ to $t_3 = 0.3$, and $t_3 = 0.3$ to $t_4 = 0.88$. Unfortunately, neither model was able to converge in the $\alpha$-shaped case.

## 4. Conclusion

We present a novel framework for modeling the dynamics of high-dimensional systems from snapshot data in a multi-marginal setting with non-equidistant time points, while remaining in the high-dimensional space and avoiding the pitfalls of dimensionality reduction. By expanding the literature of Conditional Flow Matching, we have developed a method that learns flows for overlapping triplets, enhancing robustness and stability in multi-marginal settings. We validate our method's scalability and ability to learn in high di-

mensional spaces using the CITEseq and Multiome datasets. Our application to the Imagenette dataset further demonstrates the method's generalizability to varied datasets as well as generative capabilities for sampling from a time point marginal.

The incorporation of stochasticity through score matching improves robustness and avoids overfitting, enabling the model to generalize to new conditions. This work opens new avenues for both generative algorithms and modeling cellular responses to arbitrary, user-defined perturbations, providing a computationally efficient and biologically accurate framework capable of handling the complexities of high-dimensional, stochastic biological systems.

## Acknowledgment

The authors Justin Lee and Heman Shakeri gratefully acknowledge the support provided by the University of Virginia Cancer Center. Heman Shakeri additionally thanks Capital One for their generous support of this research.

## Impact Statement

We have developed a method to robustly model the evolution over time of a system of variables given snapshot data containing samples of the system state at various arbitrary time points, scalable in both the number of variables and number of snapshots. We apply a multi-marginal optimal transport plan in order to couple samples from neighboring snapshots and use the couplings as control points to generate splines on overlapping time points which interpolate the system state. The combination of the optimal transport plan and the overlapping splines provide a more robust model less prone to overfitting whilst also minimizing the total work done to transform one snapshot to the next. One particularly important application is to help oncologists model cellular behavior and potentially accelerate the development of successful drug treatments.

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

## A. Proofs

*Proof of Theorem 2.1.* The gradient of the loss for a single interval $(t_i, t_{i+1})$ with $k$ overlapping mini-flows is given by

$$\nabla_\theta \mathcal{L}_{t_i:t_{i+1}} = \nabla_\theta \mathbb{E}\left[\left\|v_t(x;\theta) - \left(\sum_{j=0}^{k-1} u_t(x \mid z_{t_{i-j}:t_{i-j+k}})\right)\right\|^2 + (k-1)\|v_t(x;\theta)\|^2\right]$$

where the expectation is taken over $q(z_{t_{i-k+1}:t_{i+k}}), \mathcal{U}(t_i, t_{i+1})$, and $\tilde{p}(x \mid z_{t_{i-k+1}:t_{i+k}}) = \frac{1}{k}\sum_{j=0}^{k-1} p_t(x \mid z_{t_{i-j}:t_{i-j+k}})$.

We begin by considering the individual losses with respect to the mini-flows

$$\mathcal{L}_{t_i:t_{i+1}} = \sum_{j=0}^{k-1} \mathbb{E}\left[\|v_t(x;\theta) - u_t(x \mid z_{t_{i-j}:t_{i-j+k}})\|^2\right] \tag{14}$$

where the expectation is taken over $q(z_{t_{i-j}:t_{i-j+k}}), \mathcal{U}(t_i, t_{i+1})$, and $p_t(x \mid z_{t_{i-j}:t_{i-j+k}})$. We combine this into a single expectation by expanding $q$ to cover $z_{t_{i-k+1}:t_{i+k}})$, noting that the expanded window does not affect each individual expectation. Next, we combine all the mini-flow probability paths into $\tilde{p}(x \mid z_{t_{i-k+1}:t_{i+k}}) = \frac{1}{k}\sum_{j=0}^{k-1} p_t(x \mid z_{t_{i-j}:t_{i-j+k}})$ because we are functionally only sampling from a single "active" probability path at a time. The resulting combined expectation is

$$\mathcal{L}_{t_i:t_{i+1}} = \mathbb{E}\left[\sum_{j=0}^{k-1} \|v_t(x;\theta) - u_t(x \mid z_{t_{i-j}:t_{i-j+k}})\|^2\right] \tag{15}$$

taken over $q(z_{t_{i-k+1}:t_{i+k}}), \mathcal{U}(t_i, t_{i+1})$, and $\tilde{p}(x \mid z_{t_{i-k+1}:t_{i+k}})$. We derive the rest by considering the inside of the expectation as follows:

$$\sum_{j=0}^{k-1} \|v_t(x;\theta) - u_t(x \mid z_{t_{i-j}:t_{i-j+k}})\|^2 = \sum_{j=0}^{k-1} \|v_t(x;\theta)\|^2 - 2\langle v_t(x;\theta), u_t(x \mid z_{t_{i-j}:t_{i-j+k}})\rangle + \|u_t(x \mid z_{t_{i-j}:t_{i-j+k}})\|^2$$

$$= \|v_t(x;\theta)\|^2 - 2\left\langle v_t(x;\theta), \sum_{j=0}^{k-1} u_t(x \mid z_{t_{i-j}:t_{i-j+k}})\right\rangle$$

$$+ \sum_{j=0}^{k-1} \|u_t(x \mid z_{t_{i-j}:t_{i-j+k}})\|^2 + (k-1)\|v_t(x;\theta)\|^2$$

$$= \|v_t(x;\theta)\|^2 - 2\left\langle v_t(x;\theta), \sum_{j=0}^{k-1} u_t(x \mid z_{t_{i-j}:t_{i-j+k}})\right\rangle$$

$$+ \left\|\sum_{j=0}^{k-1} u_t(x \mid z_{t_{i-j}:t_{i-j+k}})\right\|^2 + (k-1)\|v_t(x;\theta)\|^2 - U$$

$$= \left\|v_t(x;\theta) - \sum_{j=0}^{k-1} u_t(x \mid z_{t_{i-j}:t_{i-j+k}})\right\|^2 + (k-1)\|v_t(x;\theta)\|^2 - U$$

where $U$ is some constant term of inner products and sums of $u_t(x \mid z_{t_{i-j}:t_{i-j+k}})$ such that $\|\sum_{j=0}^{k-1} u_t(x \mid z_{t_{i-j}:t_{i-j+k}})\|^2 = \sum_{j=0}^{k-1} \|u_t(x \mid z_{t_{i-j}:t_{i-j+k}})\|^2 + U$. We utilize the well-known fact that $\|x \pm y\|^2 = \|x\|^2 \pm 2\langle x, y\rangle + \|y\|^2$ for first, third, and fourth equalities. Crucially, in the third equality we introduce $U$ in order to "complete the square" of the sum of $u_t(x \mid z_{t_{i-j}:t_{i-j+k}})$. We use the bilinearity of the inner product in the second equality. This results in the interval loss of

$$\mathcal{L}_{t_i:t_{i+1}} = \mathbb{E}\left[\left\|v_t(x;\theta) - \sum_{j=0}^{k-1} u_t(x \mid z_{t_{i-j}:t_{i-j+k}})\right\|^2 + (k-1)\|v_t(x;\theta)\|^2 - U\right] \tag{16}$$

Finally, by taking the gradient w.r.t. $\theta$, we arrive at Theorem 2.1, noting that $U$ does not depend on $\theta$ and therefore $\nabla_\theta U = 0$.

*Proof of Corollary 2.2.* If all the mini-flow regression signals $u_t(x \mid z_{t_{i-j}:t_{i-j+k}})$ are equal, then the single interval loss recovers the CFM loss (Theorem 2.1 of Lipman et al. (2023)) up to a scalar factor.

Consider the inside of the expectation in (16) where all the $u_t(x \mid z_{t_{i-j}:t_{i-j+k}})$ are equal (denoted simply as $u_t(x \mid z)$):

$$
\left\| v_t(x;\theta) - \sum_{j=1}^{k} u_t(x \mid z) \right\|^2 + (k-1)\|v_t(x;\theta)\|^2 - U = \|v_t(x;\theta)\|^2 - 2\left\langle v_t(x;\theta), \sum_{j=1}^{k} u_t(x \mid z) \right\rangle
$$

$$
+ \left\| \sum_{j=1}^{k} u_t(x \mid z) \right\|^2 + (k-1)\|v_t(x;\theta)\|^2 - U
$$

$$
= k\|v_t(x;\theta)\|^2 - 2\left\langle v_t(x;\theta), \sum_{j=1}^{k} u_t(x \mid z) \right\rangle
$$

$$
+ \left\| \sum_{j=1}^{k} u_t(x \mid z) \right\|^2 - U
$$

$$
= k\|v_t(x;\theta)\|^2 - 2\left\langle v_t(x;\theta), \sum_{j=1}^{k} u_t(x \mid z) \right\rangle
$$

$$
+ \sum_{j=1}^{k} \|u_t(x \mid z)\|^2
$$

$$
= k\|v_t(x;\theta)\|^2 - 2k\langle v_t(x;\theta)\rangle) + k\|u_t(x \mid z)\|^2
$$

$$
= k\|v_t(x;\theta) - u_t(x \mid z)\|^2.
$$

We "complete the square" for the first and last equality, group like terms in the second and fourth equality, and use the definition of $U$ in the third equality. Once we reintroduce the expectation, we see that this is proportional to Theorem 2.1 of Lipman et al. (2023):

$$
\mathcal{L}_{t_i:t_{i+1}} = \mathbb{E}[k\|v_t(x;\theta) - u_t(x \mid z)\|^2] = k\mathbb{E}[\|v_t(x;\theta) - u_t(x \mid z)\|^2] \tag{17}
$$

## B. Cubic Splines

Cubic splines are a class of piecewise functions interpolating between control points $(t_0, x_0), \ldots, (t_n, x_n)$, taking the form

$$
S(t) = \begin{cases} S_0(t) & t_0 \le t < t_1 \\ \vdots \\ S_{n-1}(t) & t_{n-1} \le t \le t_n \end{cases}
$$

where $S_i$ is the cubic polynomial $S_i(t) = a_i(t - t_i)^3 + b_i(t - t_i)^2 + c_i(t - t_i) + d_i$ for $i$ in $0, \ldots, n-1$. There are 4 coefficients to solve for per equation which results in $n$ equations and $4n$ unknowns.

### B.1. Natural Cubic Splines

Natural cubic splines solve for the above coefficients $a_i, b_i, c_i, d_i$ by applying four conditions. The first requires the spline to interpolate the data points $(t_i, x_i)$ such that $S(t_i) = x_i$ resulting in $n+1$ constraints. The second requires $S$ to be continuous at the interior points such that $S_i(t_i) = S_{i+1}(t_i)$, resulting in $n-1$ constraints. The third and fourth conditions respectively require $S'$ and $S''$ to be continuous for a total of $2n-2$ constraints. Finally two boundary conditions are added such that $S''(t_0) = S''(t_n) = 0$. In total, we have constructed a system of equations with $4n$ unknowns and $4n$

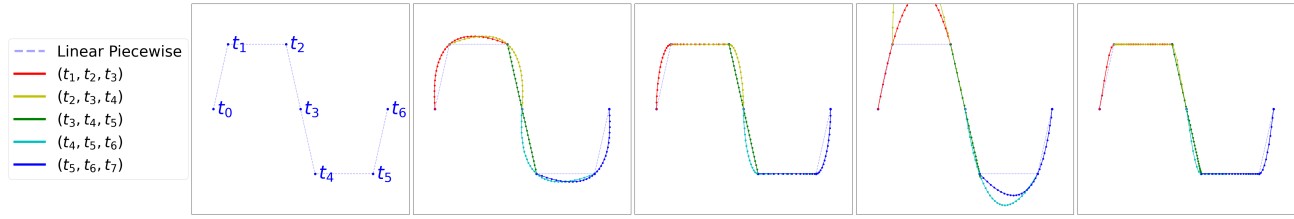

*Figure 4.* Euclidean splines on overlapping windows of size $k = 2$, using the means of each Gaussian in the S-shaped dataset as the control points. From left to right: 1) The 7 points to interpolate with time labels $t_0, \ldots, t_6$. 2) Natural cubic splines on equidistant time intervals. 3) Monotonic cubic Hermite splines on equidistant time intervals. 4) Natural cubic splines on arbitrary time intervals $\mathcal{T}_2 = (0, 0.08, 0.38, 0.42, 0.54, 0.85, 1)$. 5) Monotonic cubic Hermite splines on $\mathcal{T}_2$.

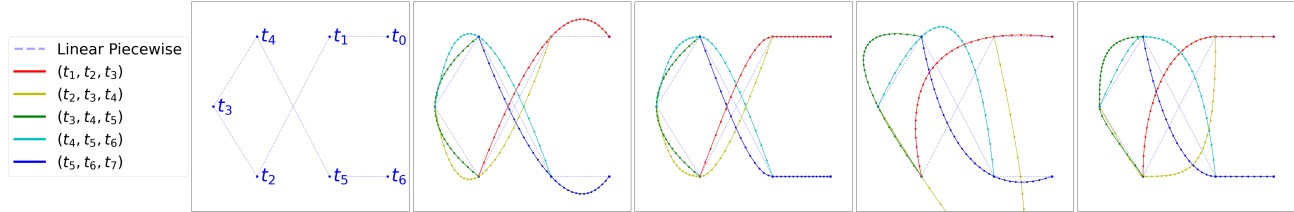

*Figure 5.* Euclidean splines on overlapping windows of size $k = 2$, using the means of each Gaussian in the $\alpha$-shaped dataset as the control points. From left to right: 1) The 7 points to interpolate with time labels $t_0, \ldots, t_6$. 2) Natural cubic splines on equidistant time intervals. 3) Monotonic cubic Hermite splines on equidistant time intervals. 4) Natural cubic splines on arbitrary time intervals $\mathcal{T}_2 = (0, 0.08, 0.38, 0.42, 0.54, 0.85, 1)$. 5) Monotonic cubic Hermite splines on $\mathcal{T}_2$.

constraints. Ultimately, this setup constructs a tridiagonal system of equations which is efficiently solvable in $\mathcal{O}(n)$ time using a single forward and backward pass. Perhaps reasonably, natural cubic splines are quite local as the influence of neighboring intervals greatly decreases the further away the neighbor is.

### B.2. Monotonic Cubic Hermite Splines

Cubic Hermite splines approach the problem differently. Consider a single time interval $[0, 1]$ and corresponding points $x_0, x_1$. Let the position of $x$ at time $t$ be given by the following cubic polynomial:

$$x_t = at^3 + bt^2 + ct + d.$$

Likewise, let $m_t$ be the velocity of $x_t$ at time $t$, given by

$$m_t = 3at^2 + 2bt + c.$$

At $t = 0$ and $t = 1$, we can solve for $x_0, x_1, m_0, m_1$ in terms of $a, b, c, d$ to get the following system of equations:

$$
\begin{aligned}
x_0 &= d \\
x_1 &= a + b + c + d \\
m_0 &= c \\
m_1 &= 3a + 2b + c.
\end{aligned}
$$

Solving this system of equations, we get

$$x(t) = (2t^3 - 3t^2 + 1)x_0 + (t^3 - 2t^2 + t)m_0 + (-2t^3 + 3t^2)x_1 + (t^3 - t^2)m_1$$

as the polynomial interpolating $(0, x_0)$ to $(1, x_1)$. All that remains is to specify values for $m_0$ and $m_1$. In other words, cubic Hermite spline algorithms are defined by how the velocities $m_i$ are selected.

Monotonic cubic Hermite splines set $m_i$ using the following strategy. Define $h_k = t_{k+1} - t_k$ and $d_k = \frac{x_{k+1} - x_k}{h_k}$. If the signs of $d_k$ and $d_{k-1}$ do not match or either is 0, then set $m_k = 0$. Otherwise, $m_k$ is given by

$$\frac{w_1 + w_2}{m_k} = \frac{w_1}{d_{k-1}} + \frac{w_2}{d_k}$$

where $w_1 = 2h_k + h_{k-1}$ and $w_2 = h_k + 2h_{k-1}$. We direct the reader to (Fritsch & Carlson, 1980) for an exact derivation and proof of monotonicity. This formula is also solvable in $\mathcal{O}(n)$ time, but differs from the natural cubic spline in that it is very local. In fact, only the immediate neighboring data points $(t_{i-1}, x_{i-1})$ and $(t_{i+1}, x_{i+1})$ influence the curve.

## C. Implementation Details

Our implementation uses NumPy (Harris et al., 2020), SciPy (Virtanen et al., 2020), and PyTorch (Paszke et al., 2017) for the mathematical operations. Plots were generated using Matplotlib (Hunter, 2007). We extended the base CFM framework provided by Tong et al. (2024a;b)

### C.1. Optimal Transport Plans

We use the Python Optimal Transport (POT) library (Flamary et al., 2021) to compute all OT plans with `ot.emd()` and the squared Euclidean distance. This method operates on two batches of samples and returns a matrix $\pi$ where the $i, j$-th entry $\pi_{i,j}$ indicates the probability of sampling $(x_{t_\ell}^{(i)}, x_{t_{\ell+1}}^{(j)}) \sim \pi(x_{t_\ell}, x_{t_{\ell+1}})$. The superscript $(i)$ denotes the $i$-th index into a batch of samples. We implement the conditional OT plan $\pi(x_{t_{\ell+1}} \mid x_{t_\ell})$ by first constructing the marginalization column vector $\hat{q}$ representing $\int \pi(x_{t_\ell}, x_{t_{\ell+1}}) dx_{t_{\ell+1}} = q(x_{t_\ell})$, where the $i$-th entry $\hat{q}_i := \sum_j \pi_{i,j}$. Finally, we construct the matrix $\hat{\pi}$ representing the conditional plan $\pi(x_{t_{\ell+1}} \mid x_{t_\ell})$ by element-wise dividing each column of $\pi$ by $\hat{q}$. Thus, the $i$-th row of $\hat{\pi}$ contains the probabilities of selecting $x_{t_{\ell+1}}^{(j)}$ given $x_{t_\ell}^{(i)}$. Because the procedure operates by coupling the indexes of samples from the data, we can consider this as an alignment operation on the initial mini-batch. Our full sampling procedure for a single mini-flow is as follows. We set the initial time index to be 0 without loss of generality with respect to any given mini-flow. All samplers sample with replacement.

---

**Algorithm 2** MMOT Sampling

Set batch size $b$
Sample batches $X_{t_0}, \ldots, X_{t_k} \sim q(x_{t_0}), \ldots, q(x_{t_k})$ where $X_{t_\ell} = \{x_{t_\ell}^{(i)}\}_{i=1}^b$
Sample $(X_{t_0}^\star, X_{t_1}^\star) \sim \pi(X_{t_0}, X_{t_1})$
**for** $\ell \in 2..k$ **do**
    Compute the conditional OT plan $\hat{\pi}(X_{t_\ell} \mid X_{t_{\ell-1}}^\star) \leftarrow \pi(X_{t_{\ell-1}}^\star, X_{t_\ell})/\hat{q}(X_{t_{\ell-1}}^\star)$
    Sample $X_{t_\ell}^\star \sim \hat{\pi}(X_{t_\ell} \mid X_{t_{\ell-1}}^\star)$
**end for**
**Output:** Aligned batches $X_{t_0}^\star, \ldots, X_{t_k}^\star$

---

### C.2. Neural ODE and SDE Solvers

We use the same setup as Tong et al. (2024b) and use the SDE solver from `torchsde` (Li et al., 2020; Kidger et al., 2021). We did not adjust any of the default hyperparameters for the solver. For the SDE model, we set the drift to $f_t(x) \leftarrow v_t(x; \theta) + \frac{g^2(t)}{2} s_t(x; \theta)$ from (12) where $v_t$ and $s_t$ are respectively the flow and score networks. The diffusion is set to a constant $g_t(x) \leftarrow \sigma = 0.15$.

## D. Experimental Setup

### D.1. Training Setup

For all experiments except for image generation, we used a MLP with an input layer, two hidden layers of width 64, an output layer, along with SELU activation functions. We optimize these networks using AdamW. We set $\sigma = 0.15$ for our method and likewise as the noise scale in MIOFlow.

For the S-shaped, $\alpha$-shaped, and DynGen datasets, we trained for 2500 gradient steps and a learning rate of 1e-4. For the CITEseq and Multiome datasets, we trained for 1000 gradient steps and a learning rate of 1e-5.

MIOFlow is a method to infer "optimal" trajectories on manifolds which correspond to geodesics. As we do not have access to the underlying manifold itself, the authors propose learning it from data using a GAE such that the encoder $\phi$ is a mapping from the ambient space to the manifold. More specifically, the encoder learns an embedding such that the Euclidean distance of two embedded points $\|\phi(x) - \phi(y)\|$ matches some geodesic distance $G(x,y)$ based on a diffusion affinity matrix.

Additionally, MIOFlow requires training a GAE to embed high-dimensional data into a lower-dimensional space and to then reconstruct trajectories learned in the embedded space. We define the encoder as a MLP with three hidden layers of sizes 128, 64, and 32. This encoder outputs an embedding into $\mathbb{R}^2$. The decoder has the same architecture but in reverse. We use ReLU as the activation function. The GAE is trained for 1000 gradient steps using the AdamW optimizer.

For Imagenette, we used the same setup as in Lipman et al. (2023) and use the U-Net architecture from Dhariwal & Nichol (2021) as well as the Adam optimizer. We resize all images to $32 \times 32$ and normalize across all RGB channels with $\mu = 0.5$ and $\sigma = 0.5$. No other preprocessing is done. We match hyperparameters where we can but opt for a smaller experiment in terms of the number of GPUs and the effective batch size per window. Note that for the Triplet model there is only one window but for the Pairwise model there are two. We reproduce the hyperparameters below.

|  | Imagenette-32 |
| --- | --- |
| Channels | 256 |
| Depth | 3 |
| Channels Multiple | 1, 2, 2, 2 |
| Head | 4 |
| Heads Channels | 64 |
| Attention Resolution | 16, 8 |
| Dropout | 0.0 |
| Effective Batch Size per Window | 192 |
| GPUs | 1 |
| Epochs | 250 |
| Iterations | 250k |
| Learning Rate | 1e-4 |
| Learning Rate Scheduler | Polynomial Decay |
| Warmup Steps | 20k |

*Table 3.* Imagenette experiment hyperparameters. We use the same model architecture and learning rate hyperparameters as Lipman et al. (2023). However, we use 250 epochs instead of 200 as we arbitrarily set an epoch to 1000 gradient steps. This matches the 250k iterations. Moreover, we have a smaller effective batch size per window of 192 and only use one GPU.

### D.1.1. SCORE MATCHING IMPLEMENTATION

As noted in Section 2.3.2, we have $\nabla \log p_t(x \mid z) = -\frac{\epsilon}{\sigma_t}$ for $\epsilon \sim \mathcal{N}(0, I)$. However, this direct formulation does not protect against numerical instability when $\sigma_t$ is small. We follow the approach used by Tong et al. (2024b) and take advantage of the user-defined weighting schedule $\lambda(t)$ to cancel out the division and learn the scaled target $\frac{g(t)^2}{2} \nabla \log p_t(x \mid z)$ based on the Fokker-Planck Equation (2). By rewriting the inside of the expectation of the scaled score loss as

$$\lambda(t)^2 \left\| \hat{s}_t(x; \theta) - \frac{g(t)^2}{2} \nabla \log p_t(x|z) \right\|^2 = \left\| \lambda(t)\hat{s}_t(x; \theta) + \lambda(t)\frac{g(t)^2 \epsilon}{2\sigma_t} \right\|^2,$$

we can see that when setting $\lambda(t) = \frac{2\sigma_t}{g(t)^2}$, the score loss becomes

$$\|\lambda(t)\hat{s}_t(x; \theta) + \epsilon\|^2 \qquad \epsilon \sim \mathcal{N}(0, I).$$

This approach allows us to reconstruct the mini-flow SDE drift as the sum of the mini-flow ODE drift and the scaled score network output:

$$u_t(x; \theta) = v_t(x; \theta) + \hat{s}_t(x; \theta). \tag{18}$$

### D.2. DynGen

We repurpose the DynGen data used in MIOFlow (Huguet et al., 2022) for our experiments. Notably, the data itself is not the raw simulated reads; it is preprocessed into 5 dimensions using PHATE (Moon et al., 2019).

PHATE operates as a dimensionality reduction scheme aiming to preserve both local and global dependency structures. Local structure is learned first by imposing Pairwise affinities under a Gaussian kernel. Global structure is inferred by propagating the local affinities via diffusion, effectively learning a statistical manifold based on the information geometry. Finally, metric MDS is used as the dimensionality reduction strategy.

We believe that the GAE used in MIOFlow learns the data manifold for the (PHATE-transformed) DynGen dataset especially well given that, by construction, the DynGen dataset does indeed reside on a manifold equipped with a diffusion-based metric. This matches the prior belief in MIOFlow that diffusion-based affinities can accurately capture the data manifold.

### D.3. CITEseq and Multiome

These datasets were published as part of a NeurIPS competition for multimodal single-cell integration (Burkhardt et al., 2022). We present a brief overview, and refer the reader to the competition itself for more in-depth descriptions [2]. The data is collected from peripheral CD34+ hematopoietic stem and progenitor cells from healthy human donors. The CITEseq data is measured using 10x Genomics Single Cell Gene Expression with Feature Barcoding technology. The Multiome data is measured using 10x Chromium Single Cell Multiome ATAC + Gene Expression technology.

Technically, both the CITEseq and Multiome datasets are labeled, with the former about predicting protein levels given gene expressions, and the latter about predicting gene expressions given ATAC-seq peak counts. We are only interested in the gene expression data, so we only use the CITEseq input data and the Multiome target data. Following Tong et al. (Tong et al., 2024b), we only select cells from the respective datasets from a single donor id `13176`. The gene expression data is already library-size normalized and log1p transformed, so we compute the PCA and top highly variable genes without any further preprocessing step.

### D.4. Imagenette

This dataset a subset of the well-known ImageNet dataset (Deng et al., 2009) containing a curated collection of 10 easily classifiable classes. The classes present are: tench, English springer, cassette player, chain saw, church, French horn, garbage truck, gas pump, golf ball, and parachute. More details and variants of the dataset can be found at the GitHub page [3].

## E. Ablation Studies

We report in Table 4 ablation experiments on held-out time points for the S-shaped and $\alpha$-shaped Gaussians on the Pairwise ($k = 1$), Triplet ($k = 2$), and "All" ($k = M - 1$) models. We test both $\mathcal{T}_1 = (0, 0.17, 0.33, 0.5, 0.67, 0.83, 1)$ and $\mathcal{T}_2 = (0, 0.08, 0.38, 0.42, 0.54, 0.85, 1)$. We evaluate the $W_1$ metric on the held-out marginal for these experiments. In general, the Pairwise model performed worst, whereas the Triplet and All models were relatively even, providing experimental validation for minimal performance boosts when $k > 2$.

## F. Flow Visualizations

We visualize our experiments here. Viewing in color is recommended.

---

[2] https://www.kaggle.com/competitions/open-problems-multimodal/overview
[3] https://github.com/fastai/imagenette

| | Held-out index | S-shaped | | | $\alpha$-shaped | | |
| --- | --- | --- | --- | --- | --- | --- | --- |
| | | Pairwise | Triplet | All | Pairwise | Triplet | All |
| $\mathcal{T}_1$ | 1 | $2.43^\dagger$ | 2.13 | **2.08** | **3.38** | 4.95 | $5.13^\dagger$ |
| | 2 | $2.59^\dagger$ | **1.96** | 2.14 | 4.38 | **4.37** | $4.84^\dagger$ |
| | 3 | $1.44^\dagger$ | 1.28 | **1.13** | **2.72** | 2.94 | $2.97^\dagger$ |
| | 4 | $2.12^\dagger$ | 1.95 | **1.74** | $3.78^\star$ | $4.54^{\star\dagger}$ | 4.53 |
| | 5 | $2.36^{\star\dagger}$ | $1.83^\star$ | 2.04 | **4.17** | $5.04^\dagger$ | 4.71 |
| $\mathcal{T}_2$ | 1 | $2.35^\dagger$ | 1.71 | **1.48** | 2.78 | 3.94 | 3.94 |
| | 2 | **2.65** | 3.27 | $3.64^\dagger$ | $5.74^\dagger$ | 4.94 | **4.68** |
| | 3 | $2.65^\dagger$ | 1.90 | **1.76** | 3.37 | **3.24** | $3.91^\dagger$ |
| | 4 | **0.86** | 1.09 | $2.11^\dagger$ | $8.08^{\star\dagger}$ | $3.79^\star$ | **2.42** |
| | 5 | $2.12^{\star\dagger}$ | $1.62^\star$ | **1.59** | 5.06 | $5.12^\dagger$ | **4.54** |

*Table 4.* $W_1$ metrics on ablation experiments varying the held-out time point. Entries with a $^\star$ indicate values reported in Table 1. Entries with a $^\dagger$ indicate the worst performance out of the Pairwise, Triplet, and All models.

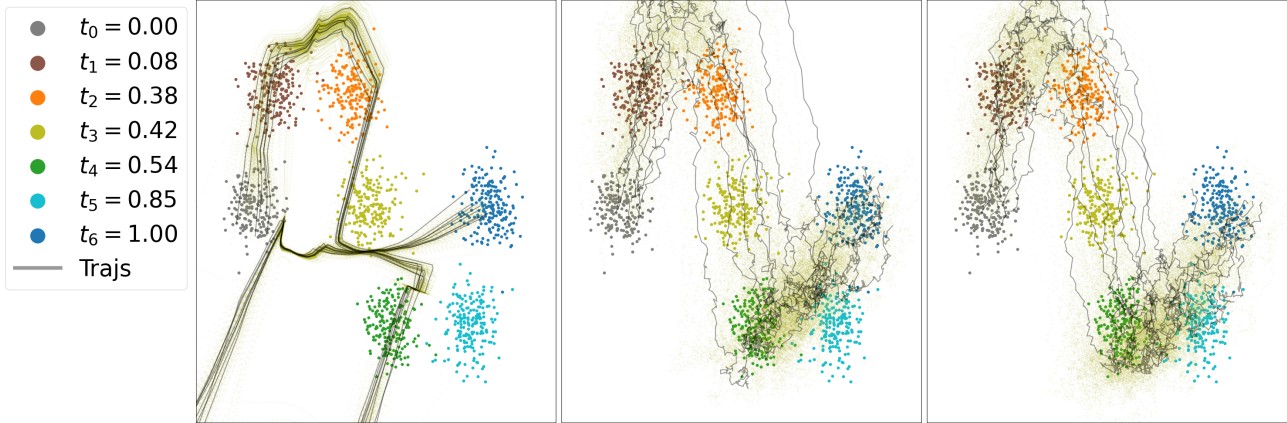

*Figure 6.* Trajectories for S-shaped Gaussians using arbitrary time points $\mathcal{T}_2 = (0, 0.08, 0.38, 0.42, 0.54, 0.85, 1)$ and holding out time point $t_5 = 0.85$. Trajectories start from the leftmost point and follow the curve to reach the rightmost point. From left to right: MIOFlow, Pairwise, Triplet.

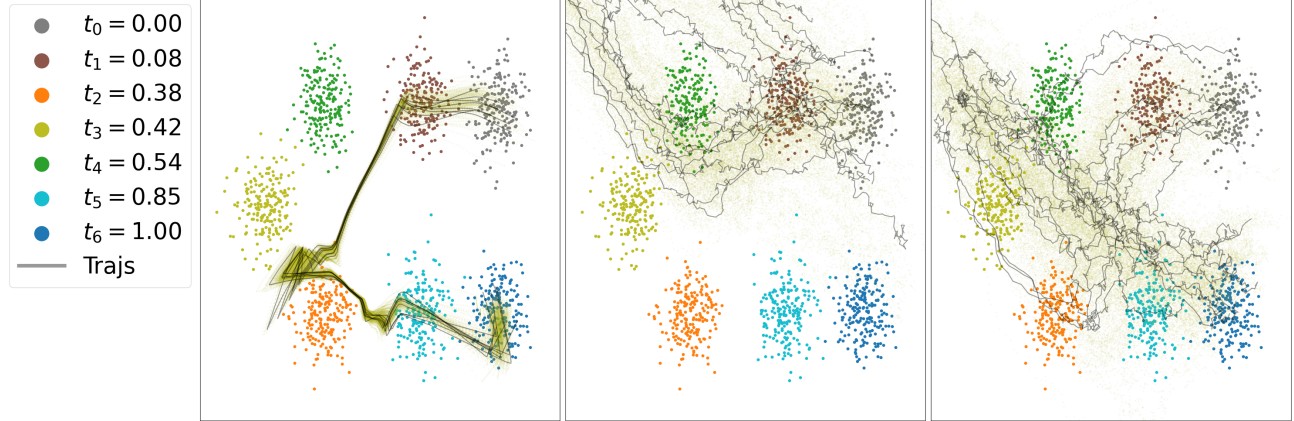

*Figure 7.* Trajectories for $\alpha$-shaped Gaussians using arbitrary time points $\mathcal{T}_2 = (0, 0.08, 0.38, 0.42, 0.54, 0.85, 1)$ and holding out time point $t_4 = 0.54$. Trajectories start from the upper right and loop around to the bottom right. From left to right: MIOFlow, Pairwise, Triplet.

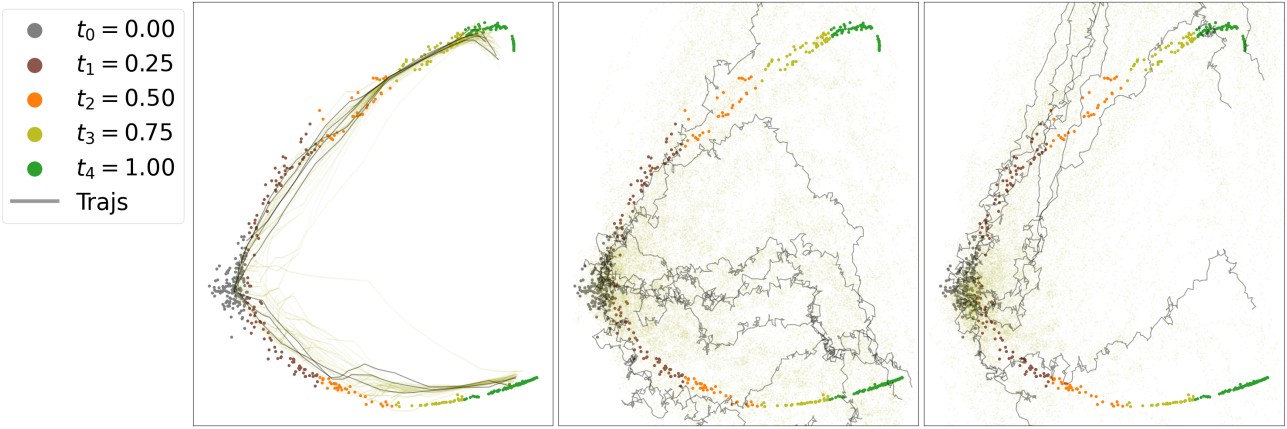

*Figure 8.* DynGen simulated trajectories. Trajectories start from the leftmost point and quickly bifurcate into the upper and lower right. The trajectories are in $\mathbb{R}^5$, but only the first and second dimensions are shown here. We hold out $t_1 = 0.25$. From left to right: MIOFlow, Pairwise, Triplet.

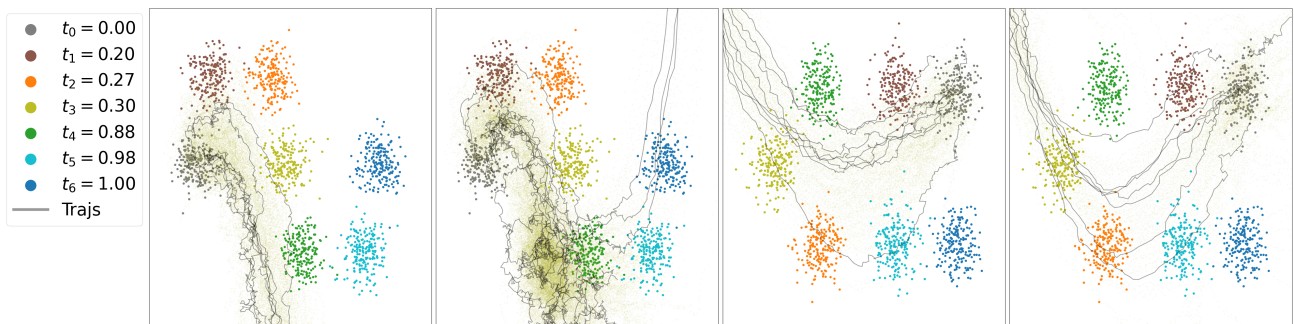

*Figure 9.* Trajectories for S and $\alpha$-shaped Gaussians predicted by the Pairwise and Triplet models using all time points from $\mathcal{T}_3 = (0, 0.2, 0.27, 0.3, 0.88, 0.98, 1)$. From left to right: 1) Pairwise on S-shaped. 2) Triplet on S-shaped. 3) Pairwise on $\alpha$-shaped. 4) Triplet on $\alpha$-shaped.

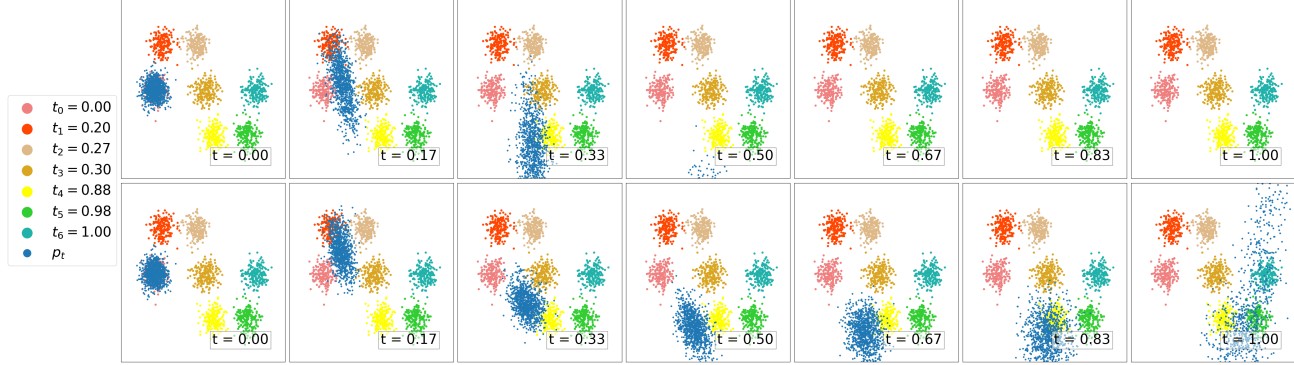

*Figure 10.* Trajectories for S-shaped Gaussians predicted by the Pairwise and Triplet models using all time points from $\mathcal{T}_3 = (0, 0.2, 0.27, 0.3, 0.88, 0.98, 1)$. Top row) Pairwise. Bottom row) Triplet.

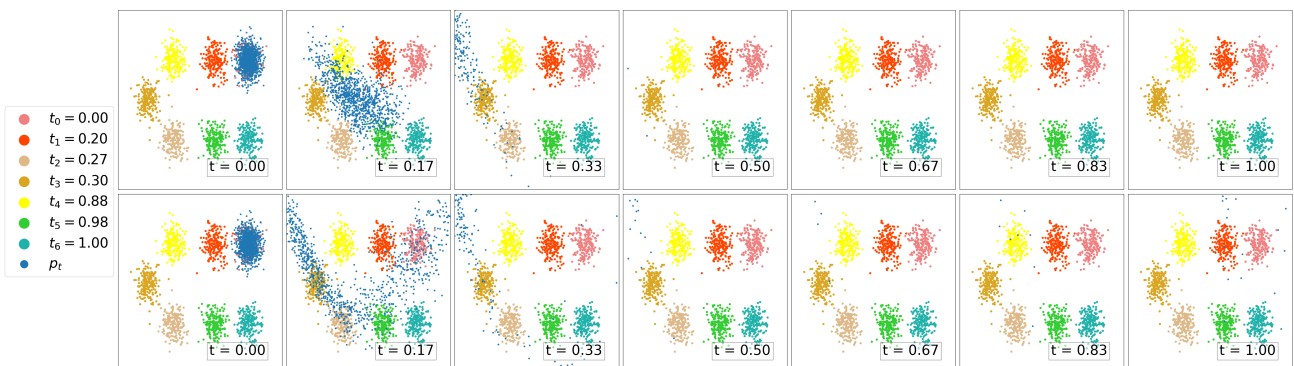

*Figure 11.* Trajectories for $\alpha$-shaped Gaussians predicted by the Pairwise and Triplet models using all time points from $\mathcal{T}_3 = (0, 0.2, 0.27, 0.3, 0.88, 0.98, 1)$. Top row) Pairwise. Bottom row) Triplet.

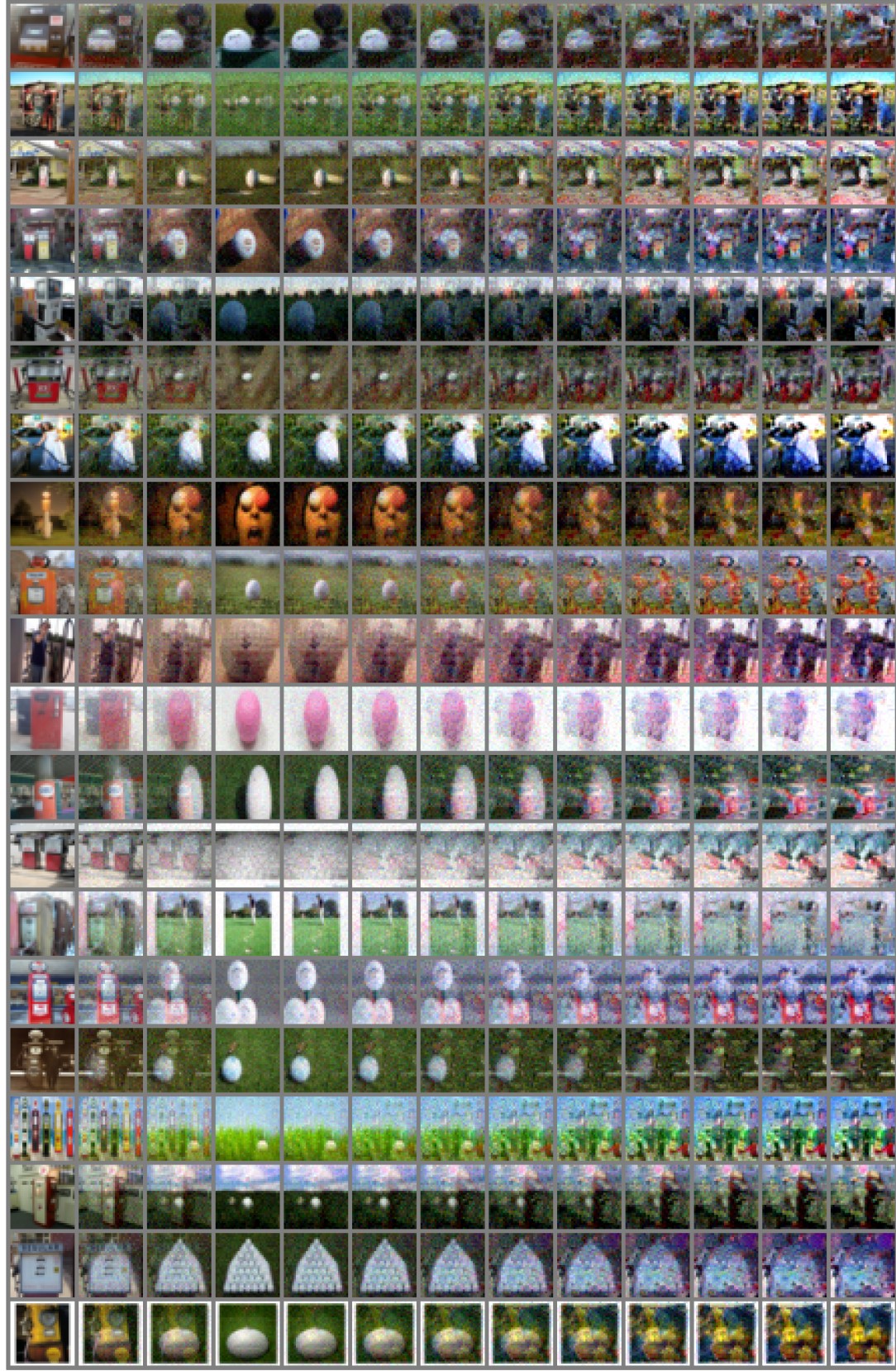

*Figure 12.* Pairwise model on Imagenette-32 for the class progression: gas pump to golf ball to parachute. Time points are from $\mathcal{T} = (0, 0.25, 1)$. Images are generated from a checkpointed model at 160k gradient steps.

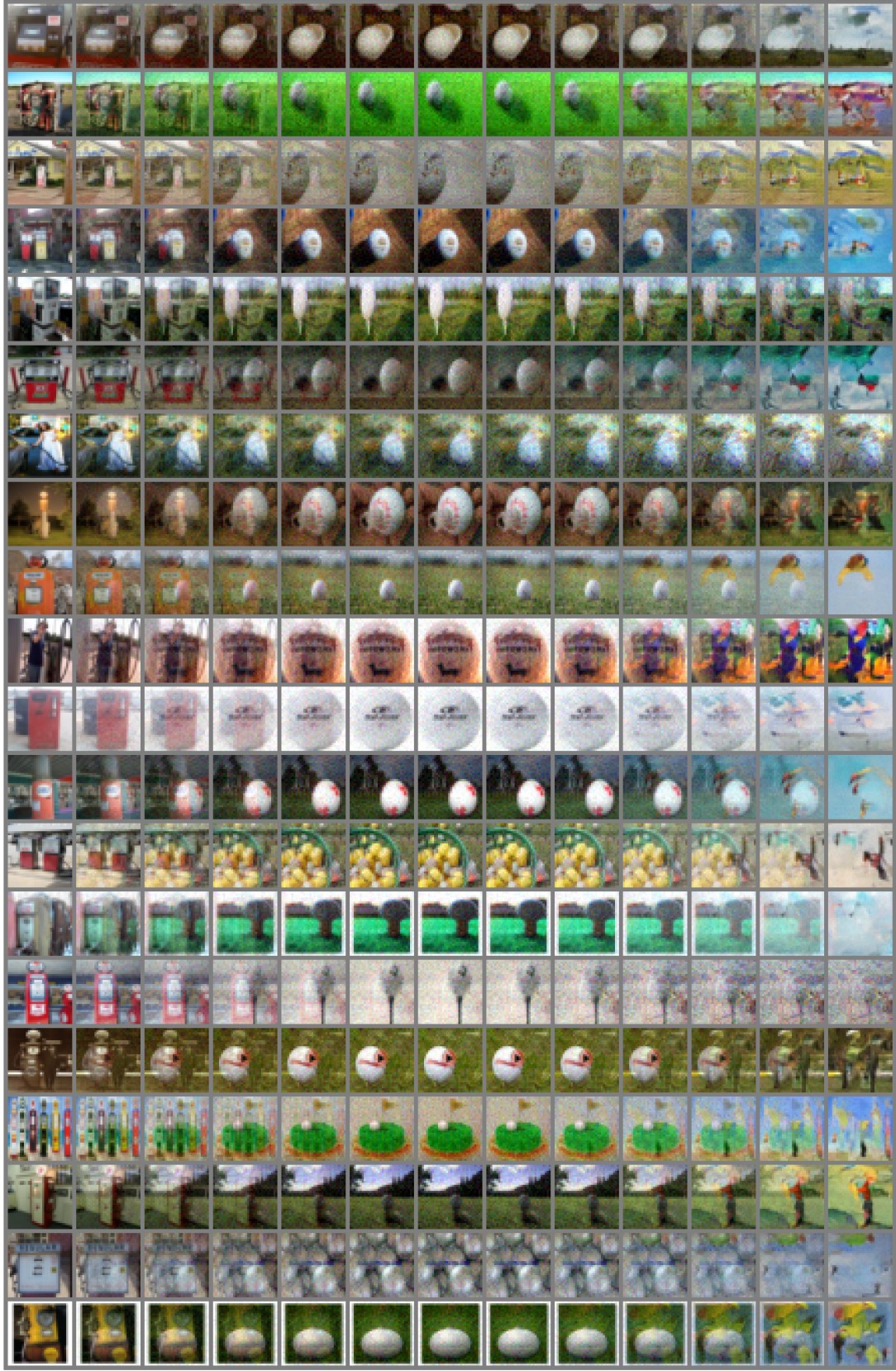

*Figure 13.* Triplet model on Imagenette-32 for the class progression: gas pump to golf ball to parachute. Time points are from $\mathcal{T} = (0, 0.5, 1)$. Images are generated from a checkpointed model at 160k gradient steps.

