# OpenReview forum: "Multi-Marginal Stochastic Flow Matching for High-Dimensional Snapshot Data at Irregular Time Points"
_ICML.cc/2025/Conference — ICML 2025 poster_

### Official Review · Reviewer_7a1W · 2025-03-12

**Overall Recommendation:** 3

**Summary:**

The authors proposed a multimarginal extension of flow matching that are simulation free and can work with high dimension data.

**Claims And Evidence:**

The theoretical claims are supported by theorem and proofs. I am a bit skeptical on the empirical performance without uncertainty.

**Essential References Not Discussed:**

I do not have one particular -- trajectory inference literature is too large.

**Experimental Designs Or Analyses:**

I checked both simulated and real data. I think the experimental design make sense.

**Methods And Evaluation Criteria:**

They make more sense if with some uncertainty.

**Other Comments Or Suggestions:**

None

**Other Strengths And Weaknesses:**

- The triplet idea is very interesting
- Scales well in dimensions.

**Questions For Authors:**

1) I want to know more is the assumptions on the marginals -- are they discrete/empirical or continuous/population?
2) related to 1), Tong et al. 2023a used this mixture of Brownian bridge results of SB problem that would produce trajectories pass through one particles at each time point, this seems not inherited in the proposed method. Is this the case and does it imply that the proposed method operates on the continuous marginals not empirical version of it?

Tong, A., Malkin, N., FATRAS, K., Atanackovic, L., Zhang, Y., Huguet, G., Wolf, G., and Bengio, Y. Simulationfree schrodinger bridges via score and flow matching. In ¨ICML Workshop on New Frontiers in Learning, Control, and Dynamical Systems, 2023a.

**Relation To Broader Scientific Literature:**

Being able to perform scalable trajectory inference is useful in biology, ecology and environmental science. And the paper is targeting at an important problem.

**Theoretical Claims:**

I did not check in depth, but did not find obvious mistakes.

---

> ### Author Rebuttal · Authors · 2025-04-01
>
> Thank you for your feedback.
>
> **Q1)**
>
> We assume that the data space of the marginals is some continuous metric space. In particular, we assume that the data lies in $R^d$ space with the standard Euclidean distance (or distance squared) $d(x, y) = \| x - y \|^2$ for all our experiments. Moreover, we also assume that all the true marginals $\rho_i$ are continuous and that our data is just an empirical realization of the $\rho_i$ marginals.
>
> **Q2)**
>
> The numerical computation is done on samples (e.g. computing OT plans with respect to samples from the marginal distributions) because we do not assume access to the true closed form functionals defining the various marginal distributions. The underpinning theory, however, is based on learning a parameterized model which can generate a time-varying probability path $p_t$ that is time-differentiable and is constrained to match the true marginals (up to the fidelity given by the data) at the respective times.
>
> In terms of trajectory inference, the SDE solvers operate on individual particles. Applying this to a point cloud outputs the trajectory of the empirical distribution, for example seen in Figure 1.
>
> [1] Tong, A., Malkin, N., FATRAS, K., Atanackovic, L., Zhang, Y., Huguet, G., Wolf, G., and Bengio, Y. Simulationfree schrodinger bridges via score and flow matching. In ¨ICML Workshop on New Frontiers in Learning, Control, and Dynamical Systems, 2023a.

---

### Official Review · Reviewer_4cti · 2025-03-13

**Overall Recommendation:** 3

**Summary:**

The paper presents Multi Marginal Stochastic Flow Matching Model(abbreviated as MMSFM), which is an extension of simulation-free score and flow matching method. The method enables the alignment of high-dimensional snapshots obtained from non-equidistant time points without reducing dimensionality. MMSFM uses third-degree polynomial to represent the non-equidistant time interval which connects optimally coupled points. Next, MMSFM introduce score matching and flow matching to reconstruct the Fokker-Planck equation while modeling stochastic process which bridges the distributions. As the non-equidistant snapshots are multi-marginal data which consists of overlapping mini-flows, MMSFM proposes a rolling window frame algorithm to deal with overlapping triplets, enhancing robustness and stability of the model. The model has been applied on synthetic datasets , single-cell dataset and other high- dimension real-world datasets, such as COLO58(melanoma single-cell) and CITEseq(Cellular Indexing of Transcriptomes and Epitopes by Sequencing)

**Claims And Evidence:**

Yes. The paper gives out several theorems to prove the feasibility of score matching and overlapping mini-flow matching. The monotonic cubic Hermite splines introduced to couple different distributions promise the monotonicity of each piece of polynomial and allow a smaller and more flexible window while guaranteeing the computational efficiency and robustness. The experiments on both synthetic datasets and real-world datasets provides a solid foundation for the method proposed in the paper.

**Essential References Not Discussed:**

No as far as I know. The paper mentions most of the related work which helps the reader understand the model.

**Experimental Designs Or Analyses:**

Yes. The paper conducts the algorithm proposed on three synthetic datasets, one single-cell dataset from COLO858 melanoma cells, and two RNA gene expression datasets. Data is collected at several non-equidistant timepoints and the results are shown in tables and illustrations in the section 3. They both proves the soundness and validity of the experimental designs.

**Methods And Evaluation Criteria:**

Yes. The algorithm MMSFM combines rolling window, optimal transportation and score matching to cope with high-dimension non-equidistant time interval snapshots. The author choose COLO58 and CITESeq as  experiment datasets, which is sampled as non-equidistant time interval distribution. The model outperforms traditional algorithms such as MIOFlow.

**Other Comments Or Suggestions:**

Some figure (such as figure 2) should be added into main body of the paper instead of showing in the supplementary material. In section 2.3, the author mentions ‘See Figure 2 for a visual representation of the variation of paths in an interval.’ While the figure 2 is only shown in the supplemental materials, which makes it difficult to read the paper.

**Other Strengths And Weaknesses:**

Strength:
1. The paper innovatively provides a novel model to solve the non-equidistant time interval snapshots inferring problem. The model based on score and flow matching outperforms traditional methods and increases the interpretability of the model.
2. The model combines rolling windows with spline measurements to address the issue of non-uniform time points which is inspired by optimal transport and Brownian bridge and shows great potential in dealing with other similar problems.
3.The experiments cover both synthetic and real-world data, verifying the capability of high-dimensional modeling, which is better than MIOFlow. Also, the visualization of the results is illustrated clearly in both section3 and supplemental materials.
Weakness:
1.The paper lacks in analysis of the bifurcation stream problem in Dyngen datasets, it should be added or be discussed in the future works.
2.The paper forgets to discuss the computational complexity of the rolling window size k in a broader view, which is an inevitable part of the theoretical analysis. The paper shows a M-dependent analysis of the computational complexity, but more analysis can be conducted in the study.

**Questions For Authors:**

1.The distant between different time intervals in the experiments shows a monotonically increasing property, what will happen if the gap between time intervals is more irregular?
2.What will happen if less snapshots is given? Is the model robust enough to solve a longer time experiment?
3. Can you explain the bifurcation of flow in Dyngen datasets and gives a analytical explanation?

**Relation To Broader Scientific Literature:**

The paper combines score and flow matching network with optimal transport and Brownian bridge to solve the multi-marginal problem which only non-equidistant time interval snapshots are provided. The newly-proposed method is one of the contribution of this work . The introduction of rolling window algorithm further improves the performance of the model and the results of experiments on both synthetic datasets and real-world datasets outperforms the traditional ones, which is another contribution of the paper.

**Theoretical Claims:**

Yes. The paper gives one theorem and one corollary. The theorem calculates the gradient of the loss for a single interval with overlapping mini-flows and the corollary claims the condition in theorem is a special case of conditional flow matching. The proof of the theorem and corollary is correct and is well shown in the supplementary material.

---

> ### Author Rebuttal · Authors · 2025-04-01
>
> Thank you for your feedback.
>
> **Q1)**
>
> This is a very insightful question, as you are correct to hint at the sensitivity of splines to the time-labels in the data.
>
> For the S-shaped and $\alpha$-shaped synthetic datasets, we evaluate on 3 different timepoint distributions to see the effect of varying the gaps.
> * Uniform gaps $T_1 = (0, 0.17, 0.33, 0.5, 0.67, 0.83, 1)$, or alternatively $(i/6)\_{i=0}^6$.
> * One set of arbitrary gaps $T_2 = (0, 0.08, 0.38, 0.42, 0.54, 0.85, 1)$. The smallest gap here is 0.04 from the interval $(0.38 \to 0.42)$. The largest gap is approximately 0.3 for which there are two intervals: $(0.08 \to 0.38)$ and $(0.54 \to 0.85)$.
> * A second set of arbitrary gaps $T_3 = (0, 0.2, 0.27, 0.3, 0.88, 0.98, 1)$. This is an extremely uneven and difficult timepoint set as it contains a disproportionately large interval of 0.58 in $(0.3 \to 0.88)$. Also included are two very small intervals of 0.02-3 in $(0.27 \to 0.3)$ and $(0.98 \to 1)$.
>
> We can see in Table 1 that for $T_1$ and $T_2$, the Triplet model tends to outperform the Pairwise model. Interestingly, the Pairwise model did outperform the Triplet model on the $\alpha$-shaped $T_1$ case, however we believe that given the Pairwise model's weakness on $\alpha$-shaped $T_2$ compared to the Triplet model, the experiments validate the ability for splines to be flexible enough for irregular timepoints.
>
> Still, it is probably a good idea to be cautious. We can see snapshots of the probability path given in the $T_3$ case by plotting how the point cloud moves in time. The relevant figures are in Appendix E, Figures (9, 10). In the S-shaped case (Figure 9) we see that the Triplet model learns the flow but includes shearing effects. Moreover, in the $\alpha$-shaped case (Figure 10) neither model was able to successfully learn the flow. We believe exploring regulation effects to prevent this behavior can lead to fruitful future work.
>
> The COLO858 dataset contains 8 timepoints of $T = (0, 0.5, 2, 6, 15, 24, 72, 120)$. We normalize this to $T = (0, 0.004, 0.017, 0.05, 0.125, 0.2, 0.6, 1)$. We can see that the size of the gaps is monotonically increasing. Table 1 again shows the Triplet model to outperform the Pairwise model in this setting with irregular timepoints.
>
> The Multiome and CITEseq datasets both contain 4 timepoints of $T = (2, 3, 4, 7)$ which we normalize to $T = (0, 0.2, 0.4, 1)$. Likewise, we see that the gap size is monotonically increasing. Table 2 shows that Triplet model marginally outperforms the Pairwise model.
>
> We see that the time intervals are monotonically increasing for COLO858, Multiome, and CITEseq because typically cell dynamics are most active early into the perturbation and gradually slow down over time. The higher time-fidelity in the early time points reflects this.
>
> Our experimental setup descriptions can be found in section 3.1.
>
> **Q2**
>
> Flow matching methods including ours are ultimately attempts at learning the dynamics driving a system subject to constraints that the marginals at given times must match certain distributions. Moreover, these underlying dynamics are not necessarily unique, implying that there are infinitely many dynamics which solve the system subject to the marginal constraints. It is this unidentifiability which leads prior work to assume linear interpolations and our work to assume spline interpolations.
>
> We are also beholden to problems relating to signal sampling which are beyond the scope of this work. Consider, for example, that we have data generated from a true underlying process of $f(t) = \sin(t)$. If we happen to take measurements at $t = 0, \pi, 2\pi, \dots, n\pi$ then our signal is a constant 0. Based on the data $\{ (k\pi, 0) \}_{k=0}^n$, the simplest reconstruction without any additional assumptions on the data or underlying process is $\tilde{f}(t) = 0$ but in fact $\tilde{f} \neq f$!
>
> With this in mind, decreasing the number of snapshots is equivalent to decreasing the time-fidelity of our data. As such, this would increase the "simplicity" of the dynamics learned by our model. On the other hand, the total length of the process is not as important given that we normalize time to run for $t \in [0, 1]$.
>
> **Q3**
>
> Yes, the Dyngen dataset introduces a bifurcation as can be seen in Appendix E, Figure 7. We repurpose the synthetic dataset from [1], which the authors generated using the Dyngen simulator [2]. We have used the same data files as [1] which defines a realistic dynamic cellular process.
>
> **References**
>
> [1] Huguet, G., Magruder, D. S., Tong, A., Fasina, O., Kuchroo, M., Wolf, G., and Krishnaswamy, S. Manifold interpolating optimal-transport flows for trajectory inference. Advances in neural information processing systems, 35: 29705–29718, 2022.
>
> [2] Cannoodt, R., Saelens, W., Deconinck, L., and Saeys, Y. Spearheading future omics analyses using dyngen, a multi-modal simulator of single cells. Nature Communications, 12(1):3942, 2021.

---

### Official Review · Reviewer_cNq2 · 2025-03-15

**Overall Recommendation:** 4

**Summary:**

This paper proposes an extension of flow-matching for multi-marginals - i.e. when multiple snapshots are observed, typically over time. The method sample conditioning points from all snapshots using an approximation of the multi-marginal optimal transport map and then fits a spline to these points that is used as the conditional drift. The authors then evaluated their method on synthetic and single-cell data.

**Claims And Evidence:**

The authors show in their experiments (both synthetic and single-cell) that the multi-marginal extension is more effective than pairwise flow matching. This demonstrates the added value of their approach.

**Essential References Not Discussed:**

The authors missed an important, albeit very recent, reference that is seemingly very related to their approach [1]. Given the papers are concomitant, I don't expect the authors to compare against it but I encourage them to position their paper with respect to it.

The paper is also related to [2]. The authors could compare their method against it.


[1] Rohbeck, Martin, et al. "Modeling Complex System Dynamics with Flow Matching Across Time and Conditions." The Thirteenth International Conference on Learning Representations, 2025.
[2] Sinho Chewi, Julien Clancy, Thibaut Le Gouic, Philippe Rigollet, George Stepaniants, and Austin Stromme. Fast and smooth interpolation on Wasserstein space. In International Conference on Artificial Intelligence and Statistics, pp. 3061–3069. PMLR, 2021.

**Experimental Designs Or Analyses:**

The experimental designs are sound, the authors learnt their method on given snapshots and evaluated on a left out time point, which is typically done in the literature.

**Methods And Evaluation Criteria:**

The evaluation criteria are clear and make sense (reconstruction of held-out snapshots).

**Other Comments Or Suggestions:**

Cfr above.

**Other Strengths And Weaknesses:**

This paper addresses an important use case of flow matching, where multiple snapshots are observed over time. The authors proposed a sound and effective way to leverage temporal dependencies and showed favorable experimental results.

Weaknesses:

- The authors claim to propose a promising method for single-cell perturbation but I did not see any mechanism to incorporate perturbations nor any experimental result involving perturbations.

- The method heavily relies on the assumption that the temporal dynamics follow a spline. It's not clear whether this assumption is justified in practice.

**Questions For Authors:**

1. The authors claim to propose a promising method for single-cell perturbation but I did not see any mechanism to incorporate perturbations nor any experimental result involving perturbations. Could you please clarify the claim ?

2. The method heavily relies on the assumption that the temporal dynamics follow a spline. It's not clear whether this assumption is justified in practice. Can the authors comment on that ?

3. Could the authors include [2]  as a baseline ?



[2] Sinho Chewi, Julien Clancy, Thibaut Le Gouic, Philippe Rigollet, George Stepaniants, and Austin Stromme. Fast and smooth interpolation on Wasserstein space. In International Conference on Artificial Intelligence and Statistics, pp. 3061–3069. PMLR, 2021.

**Relation To Broader Scientific Literature:**

The paper extends the flow matching framework, and embeds itself in that literature.

**Theoretical Claims:**

I did not check all theoretical claims but I did not spot any major issue.

---

> ### Author Rebuttal · Authors · 2025-04-01
>
> Thank you for your thoughtful feedback.
>
> **Q1)**
>
> Thank you for the clarifying question. By perturbations we mean a system which is not currently at a steady state. For example, a cell system can be perturbed by some drug stimuli. Further, we focus on cases where the perturbation is fixed. Learning a model to handle user-defined, arbitrary perturbation cases (e.g. arbitrary drug types and dosages) is an important avenue for future work.
>
> **Q2)**
>
> The original flow matching method assumes a linear interpolation function to fill in the gaps for particles at times not given in the training data. This is based on the fact that we effectively do not have access to the true temporal dynamics---if we did, there would be no reason to use a flow matching framework because we could then directly simulate the system for a desired initial condition. As such, we opt for an Occam's razor approach about using simpler dynamics. In our case, this amounts to cubic splines which offer tractable solutions and avoid Runge's phenomenon. As you correctly point out, our method assumes the temporal dynamics to be approximable or model-able using spline interpolations in the data space. These data-space interpolations are then used to learn a vector field generating the probability path $p_t$. This $p_t$ is also a spline, but one over measure space, interpolating between the marginal distributions $\rho_i$ given in the data.
>
> **Q3)**
>
> Our sampling algorithm is the Transport Spline Interpolation in Algorithm 1 of [2]. However, those authors look at a single interpolating spline whereas we generate multiple overlapping splines in our own Algorithm 1 via a rolling window over the timepoint marginals. Moreover, we require a generative model for evaluations on unseen initial conditions which [2] does not provide because it looks only at interpolations between existing points.
>
> **Comments**
>
> Thank you for bringing [1] to our attention as we had not seen it at the time of developing our model. Certainly, there is a good amount of similarity and relevancy:
>
> a) We both use splines as interpolants.
>
> b) We both arrive at the same MMOT plan approximation in our work's equation (8). The authors of [1] discuss this in their Appendix B.
>
> c) We both allow for irregular snapshot timings.
>
> However, we differ in that:
>
> a) We use splines on overlapping triplets whereas [1] looks at splines over the whole sequence.
>
> b) We explore the spline algorithm and opt for monotonic Hermite cubic splines whereas [1] opts for natural cubic splines. The latter is perhaps nicer analytically, but we nonetheless use the former due to practical concerns involving overshoot. For example, natural cubic splines enforce $C^2$ continuity which produce "smoother" splines, but this constraint can introduce severe overshooting which we illustrate in Figures (2, 3, 4) found in our Appendix B. Monotonic Hermite cubic splines only enforce $C^1$ continuity which is sufficient in terms of the general Flow Matching framework. This weaker smoothness constraint, along with the monotonicity between control points, avoids the overshooting problem whilst also still allowing for continuous derivatives at each interior point unlike a simple linear interpolation.
>
> c) Related to (b), we explore the sensitivity of our method to highly irregular timepoints. This is in part because we suspect the overshooting behavior in the natural cubic splines to be caused by neighboring short and long time intervals. For example, consider our timepoint set $T_3 = 0, 0.2, 0.27, 0.3, 0.88, 0.98, 1)$ for synthetic data. Notice that the sequence $0.27 \to 0.3 \to 0.88$ involves a short interval of 0.03 followed by a long interval of 0.58. This means that any change in velocity and acceleration along the short interval can happen relatively quickly, but the corresponding change for the long interval must be drawn out. The continuity of the acceleration does not help in this regard as it prevents the spline from instantaneously re-adjust velocities.
>
> **References**
>
> [1] Rohbeck, Martin, et al. "Modeling Complex System Dynamics with Flow Matching Across Time and Conditions." The Thirteenth International Conference on Learning Representations, 2025.
>
> [2] Sinho Chewi, Julien Clancy, Thibaut Le Gouic, Philippe Rigollet, George Stepaniants, and Austin Stromme. Fast and smooth interpolation on Wasserstein space. In International Conference on Artificial Intelligence and Statistics, pp. 3061–3069. PMLR, 2021.

---

> > ### Comment · Reviewer_cNq2 · 2025-04-04
> >
> > Thank you for your answers. I confirm my score.
> >
> > - I appreciate the in-depth discussion of the method with [1]. Please make sure to include part of this in your final manuscript as this will help readers understand the subtleties of both approaches.
> >
> > - Regarding to [2], the authors in [1] did compare against it so I assume you should be able to do the same ?
> >
> > Best Regards,

---

### Official Review · Reviewer_rnGo · 2025-03-20

**Overall Recommendation:** 2

**Summary:**

This work proposes Multi-Marginal Stochastic Flow Matching (MMSFM) with the goal of training a translation model across multiple snapshots taken at non-equidistant time points. MMSFM builds upon the Flow Matching framework and extends it through measure-valued splines. (Stochastic) Flow Matching can be applied to train a pair-wise flow between all adjacent time point snapshots. However, this approach can struggle to capture the global dynamics across multiple time points and does not generalize well to arbitrary held-out time points. MMSFM enables capturing local dynamics across uneven time intervals and maintaining consistency between overlapping windows.

**Claims And Evidence:**

The authors claim that "the use of measure-valued splines enhances robustness to irregular snapshot timing, and score matching prevents overfitting in high-dimensional spaces." While this is somewhat supported by empirical evidence, this overstates the contribution of the paper as the use of the score-matching loss is taken from prior work.

Also, the authors propose using mini-batch OT, following previous work. However, no ablation of this design choice is done, and it remains unclear how reliant MMSFM is on it.

Empirically, adding splines with triplets (k=2) compared to pairwise (k=1) shows overall improvements, although not consistent across different experimental settings.

**Essential References Not Discussed:**

[3] is very closely related to MMSFM and related work as it also tackles unpaired single-cell translation. It improves upon mini-batch OT sampling through unbalanced Optimal Transport. These ideas could in future also be applied to MMSFM and would additionally be a relevant competing method.

[3] Luca Eyring, Dominik Klein, Théo Uscidda, Giovanni Palla, Niki Kilbertus, Zeynep Akata, Fabian Theis. "Unbalancedness in Neural Monge Maps Improves Unpaired Domain Translation". ICLR 2024.

**Experimental Designs Or Analyses:**

Yes, they do make sense. The synthetic datasets make sense and the biological applications demonstrate practical relevance. However, the work is submitted under "**Primary Area:** Applications->Health / Medicine" while having no modality specific metric. Measuring W1/W2 and MMD is fine but I think these experiments would be strengthened through biologically meaningful metrics as e.g. leveraged in [3].

[3] Luca Eyring, Dominik Klein, Théo Uscidda, Giovanni Palla, Niki Kilbertus, Zeynep Akata, Fabian Theis. "Unbalancedness in Neural Monge Maps Improves Unpaired Domain Translation". ICLR 2024.

**Methods And Evaluation Criteria:**

The evaluation approach is appropriate. The W1/W2 metrics provide standard measures of distribution similarity, and MMD metrics offer complementary insights. The synthetic datasets test specific capabilities (bifurcations, topology changes), and the biological applications demonstrate practical relevance. Biological metrics could further strengthen this experimental section.

Empirically, the authors compare MMSFM (k=2) to SF2M and MIOFlow. It could be beneficial to include more competing works here, as well as include results for k > 2 to shed further insight into the parameter k. While Appendix D gives one ablation on this, it is not at all discussed in the main text. Additionally, it remains unclear how e.g. k=3 would perform and what tradeoff choosing larger k would give w.r.t performance and computational cost.

**Other Comments Or Suggestions:**

Formally introduce multi-marginal optimal transport. A toy-example visualization could strengthen the clarity of the work. In general, a concrete description of all used hyperparameters seems to be missing.

**Other Strengths And Weaknesses:**

Clarity could be improved in Section 2. Specifically, regarding the term MMOT.

**Questions For Authors:**

- What solver and hyperparameters do you use to compute the OT plans in Equation 8?
- What is here referred to as MMOT (Multi-marginal OT) is implemented by computing pair-wise OT plans across adjacent time points. Could the authors extend the sentence "Then, we compute the MMOT plan given by the first-order Markov approximation" and explain how this is an MMOT plan?
- Missing description and ablation of hyperparameters for SDE solver, which solver is used, and how many NFE are used? How sensitive is the trained model to the NFEs?
- Does MMSFM also work without min-batch OT? How much worse does it perform?

**Relation To Broader Scientific Literature:**

This paper proposes to improve existing Flow/Bridge Matching literature for mapping across multiple non-equidistant time points. Prior work has shown that these methods are competitive in this problem. Through the addition of measure-valued splines, MMSFM achieves improvements in this specific problem and takes a step toward solving this problem.

**Theoretical Claims:**

The theoretical foundation draws from established work on optimal transport and measure-valued splines. The paper states that their approach minimizes total action across time points but doesn't prove that the overlapping window strategy actually achieves this global minimization.

 Additionally, I am unsure about the author's use of the term MMOT (Multi-marginal Optimal Transport). This term, see e.g. [1], is usually used in a different context with the goal of obtaining a **joint** coupling across N different marginals. This can cause confusion, as this is also the notion that is thoroughly discussed in [2]. As also mentioned by the authors, [2] tackles the multi-marginal OT problem, i.e., learning a mapping between **all pairs** of distributions. On the contrary, MMSFM learns a sequential mapping across time-adjacent distributions, which is not MMOT. Adding a formal definition of MMOT following e.g. [1], and building upon that would help clarify this confusion and position MMSFM better within related work.

[1] Brendan Pass. "Multi-marginal optimal transport: theory and applications". 2014.

[2]Michael S. Albergo, Nicholas M. Boffi, Michael Lindsey, Eric Vanden-Eijnden. "Multimarginal generative modeling with stochastic interpolants". 2023.

---

> ### Author Rebuttal · Authors · 2025-04-01
>
> Thank you for your thoughtful feedback.
>
> **Q1)**
>
> We use the Python Optimal Transport (POT) package for computing the OT plans $\pi(x_0, x_1)$ where we use $x_0, x_1$ as notational shorthand for $x_{t_\ell}, x_{t_{\ell+1}}$. The conditional plans are generated by using probability rules where $\pi(x_{\ell+1} | x_\ell) = \pi(x_\ell, x_{\ell+1}) / q(x_\ell)$. Specifically, we use the Earth Movers Distance function (`POT.emd()`) with exact matching.
>
> We manipulate the probability matrices $\pi$ returned by `POT.emd()` representing $\pi(x_i, x_{i+1})$ by using the column vector $\hat{q}$ representing $\int \pi(x_i, x_{i+1})dx_{i+1}$. We construct $\hat{q}$ by summing over the columns of $\pi$. We can then obtain the conditional plan $\hat{\pi} \gets \pi(x_{i+1} | x_i) = \pi / \hat{q}$.
>
> Sampling itself is at simple as using $\pi$ as a 2D probability table or $\hat{\pi}$ as a list of 1D probability vectors where the row index corresponds to the conditioning variable. We also note that the whole sampling procedure technically operates on the indexes of a mini-batch and so we can also consider this as an alignment operation on the initial mini-batch. Further, this procedure adopts Algorithm 1 from [2] using OT plans in place of OT maps.
>
> **Q2)**
>
> As you sharply note, the "true" MMOT plan would correspond to the joint distribution $\pi$ which considers all pairs of distributions. However, this is not easy to compute and moreover we are specifically interested in processes which do have a temporal ordering. With this in mind, we turn to the framework of joint distributions and apply the chain rule of probability to obtain that $\pi(x_0, \dots, x_M) = \pi(x_0, x_1) \prod_{i=2}^M \pi(x_i | x_{< i})$. Finally, by considering a sequential sampling process where $x_i$ must be sampled prior to $x_j$ for $i < j$, we can apply a first-order Markov approximation where $x_i$ only depends on $x_{i-1}$. Thus, the conditional probabilities in the product all reduce to the form $\pi(x_i | x_{i-1})$ and we recover equation (8). It is in this sense that we call (8) the first-order Markov approximation to the true MMOT plan.
>
> **Q3)**
>
> We use the same setup as [3] and use the SDE solver `torchsde` from `github.com/google-research/torchsde`. The drift function $f_t(x)$ is set to the SDE drift $u_t(x ; \theta) = v_t(x ; \theta) + \frac{g^2(t)}{2} s_t(x ; \theta)$ from equation (12) for some learned deterministic flow $v_t$ and score $s_t$. We set the diffusion schedule $g(t) = \sigma$ for a constant $\sigma = 0.15$.
>
> We have not explored any ablation to the SDE solver hyperparameters nor the sensitivity to the number of function evaluations, however we believe these to be avenues for interesting future studies. If referring to differences between trajectories inferred by an SDE and an ODE, neither meaningfully differed from one another because the the same learned flow model $v_t$ is used in both cases.
>
> **Q4)**
>
> Yes, MMSFM can work without mini-batch OT. Notice how in our Theorem 2.1 we define the regression signal to be certain $u_t(x | z)$ objects with an expectation over some $q(z)$. OT comes into play when constructing $q(z) \gets \pi(z)$ for sampling $z = (x_0, \dots, x_M)$ and evaluating the expectation. However, the only strict requirement is that $\pi(z)$ be a coupling distribution such that $q(x_i) = \int \pi(x_0, \dots, x_M)dx_{-i}$. We can just as easily construct $\pi(z) = \prod_i^M q(x_i)$ and satisfy the requirement, noting that this $\pi$ is not the OT plan. Based on prior work such as [1], we focused on the OT coupling and did not evaluate on the independent coupling.
>
> **Comments**
>
> Thank you for suggesting [4]. Although we have not applied the methodologies and metrics referenced, we find them very relevant and valuable for future work.
>
> **References**
>
> [1] Tong, A., Malkin, N., Huguet, G., Zhang, Y., Rector-Brooks, J., Fatras, K., Wolf, G., and Bengio, Y. Improving and generalizing flow-based generative models with mini-batch optimal transport. arXiv preprint arXiv:2302.00482, 2023b.
>
> [2] Chewi, S., Clancy, J., Le Gouic, T., Rigollet, P., Stepaniants, G., and Stromme, A. Fast and smooth interpolation on wasserstein space. In International Conference on Artificial Intelligence and Statistics, pp. 3061–3069. PMLR, 2021.
>
> [3] Tong, A., Malkin, N., FATRAS, K., Atanackovic, L., Zhang, Y., Huguet, G., Wolf, G., and Bengio, Y. Simulation-free schrodinger bridges via score and flow matching. In ICML Workshop on New Frontiers in Learning, Control, and Dynamical Systems, 2023a.
>
> [4] Luca Eyring, Dominik Klein, Théo Uscidda, Giovanni Palla, Niki Kilbertus, Zeynep Akata, Fabian Theis. "Unbalancedness in Neural Monge Maps Improves Unpaired Domain Translation". ICLR 2024.

---

> > ### Comment · Reviewer_rnGo · 2025-04-08
> >
> > Thank you for the explanations.
> >
> > **Re Q1):**
> >
> > I am not super familiar with the POT library, but EMD would suggest you are computing the Wasserstein-1 distance? Is that correct?
> >
> > **Re Q2):**
> >
> > "However, this is not easy to compute and moreover we are specifically interested in processes which do have a temporal ordering."
> >
> > This is definitely true, but I'm still very unsure whether it is right to use the term MMOT here. I would suggest to either change this or provide a formal definition of MMOT, and build upon that. In its current form claiming that MMSFM is learning MMOT is not correct in my opinion.
> >
> > **Re Q3):**
> >
> > How many NFEs are used for MMSFM during inference? The use of mini-batch OT with splines could mean that fewer NFE are needed in MMSFM compared to other methods. An ablation on the NFE would be interesting here. Additionally, I believe it is quite important to include these details as well as the details of how you are computing the OT plans into the paper/Appendix.
> >
> > **Re Q4:**
> >
> > An ablation on this could help further verify the effectiveness of the proposed approach (no mini-batch OT, but still using splines).
> >
> >
> > I choose to retain my score for now as I still think the usage of the term MMOT is not really accurate and that the explanation and introduction of MMOT needs to be improved to avoid confusion.

---

### Decision · Program_Chairs · 2025-05-01

**Decision:**

Accept (poster)

**Comment:**

This paper considers the problem of  modeling high-dimensional system dynamics from irregularly sampled snapshot data. The paper introduces multi-marginal stochastic flow matching, which extends simulation-free score and flow matching methods to the multi-marginal setting. The paper demonstrates the approach on synthetic benchmarks and single-cell gene expression data.

Overall, the reviews for this paper are positive, and the majority of reviewers were in favor of accepting the paper. The paper is well-written and has a strong application-focused motivation, while proposing novel methodology that uses simulation-free approaches to address to multi-marginal problem.

In the next revision of the paper, we encourage the authors to address Reviewer rnGo's comment about clarifying the term MMOT and other contexts of the term in the literature. Additionally, please incorporate the references discussed with reviewers during the rebuttal and other changes suggested by the reviewers.